# Dynamic Pricing in the Linear Valuation Model using Shape Constraints

**Daniele Bracale**[*]                                                    *dbracale@umich.edu*
*Department of Statistics*
*University of Michigan*

**Moulinath Banerjee**                                              *moulib@umich.edu*
*Department of Statistics*
*University of Michigan*

**Yuekai Sun**                                                          *yuekai@umich.edu*
*Department of Statistics*
*University of Michigan*

**Kevin Stoll**                                                        *kstoll@welltower.com*
*Welltower Inc.*

**Salam Turki**                                                      *STurki@welltower.com*
*Welltower Inc.*

**Reviewed on OpenReview:** *https: // openreview. net/ forum? id= uKZOR4IQaO*

## Abstract

We propose a shape-constrained approach to dynamic pricing for censored data in the linear valuation model eliminating the need for tuning parameters commonly required by existing methods. Previous works have addressed the challenge of unknown market noise distribution $F_0$ using strategies ranging from kernel methods to reinforcement learning algorithms, such as bandit techniques and upper confidence bounds (UCB), under the assumption that $F_0$ satisfies Lipschitz (or stronger) conditions. In contrast, our method relies on isotonic regression under the weaker assumption that $F_0$ is $\alpha$-Hölder continuous for some $\alpha \in (0, 1]$, for which we derive a regret upper bound. Simulations and experiments with real-world data obtained by Welltower Inc (a major healthcare Real Estate Investment Trust) consistently demonstrate that our method attains lower empirical regret in comparison to several existing methods in the literature while offering the advantage of being tuning-parameter free.

## 1 Introduction

Dynamic pricing is the process of continuously adjusting product prices in response to customer feedback based on statistical learning and policy optimization. As a fundamental aspect of revenue management, dynamic pricing has been widely applied across various industries. A key challenge in this area is balancing the need to explore customer demand with exploiting current knowledge to set optimal prices that maximize revenue. This tradeoff between exploration and exploitation has been extensively studied in fields such as statistics, machine learning, economics, and operations research Besbes & Zeevi (2009); Keskin & Zeevi (2014); Cheung et al. (2017); Cesa-Bianchi et al. (2019); den Boer & Keskin (2020). A large literature focuses on an important dynamic pricing problem where contextual information, such as product features and market conditions, is available at each time step. By leveraging this contextual data, we aim to refine pricing strategies and improve revenue outcomes. This approach, known as feature-based or contextual

---

[*]Corresponding author

pricing, allows for more customized pricing decisions that better reflect product heterogeneity, leading to more effective revenue management in today's data-rich environment.

This paper focuses on the problem of pricing a single product over a finite time horizon $T$, where the market value $v_t$ of the product is unknown to the seller and may vary over time $t = 1, 2, \ldots, T$. The market value is modeled as a linear function of observed features (covariates) of the product

$$v_t = \theta_0^\top x_t + z_t, \tag{1}$$

where $x_t \in \mathbb{R}^d$ contains 1 in the first component to consider for the intercept, $\theta_0$ is some unknown parameter and $x_t$ are independent of the noise $z_t$ that are i.i.d. with unknown cumulative distribution function (c.d.f.) $F_0$. After the seller proposes a price $p_t = p_t(x_t)$, they observe whether the item is sold or not, i.e. $y_t = \mathbf{1}\{p_t \leq v_t\}$ and collect revenue $p_t y_t$. The seller aims to design a policy that maximizes the total revenue $\sum_{t=1}^{T} p_t y_t$, given the uncertainty in the market value and the limited information available to the seller, that is $(p_t, x_t, y_t)$. The determination of the optimal revenue entails learning the model parameters $(\theta_0, F_0)$ for which various statistical tools have been employed such as kernel-based methods, bandit technique, and UCB (Fan et al., 2021; Luo et al., 2022; 2024; Xu & Wang, 2022; Tullii et al., 2024).

Building on the semi-parametric structure of the model and recent advances in shape-constrained statistics, we propose a novel policy that requires minimal assumptions about the underlying distribution of market noise. Specifically, we estimate $\theta_0$ using ordinary least squares (OLS) and $F_0$ using non-parametric least squares (NPLS), subject to the natural constraint that $F_0$ is non-decreasing.

A key advantage of our shape-constrained approach is that it is entirely data-driven and does not require the specification of any tuning parameters, unlike existing non-parametric methods. For example, the kernel-based technique proposed by Fan et al. (2021) requires bandwidth selection for optimizing the error of the estimator. In contrast, the UCB-based strategy of Luo et al. (2022) requires a set of subjective parameters including a tuning parameter.

**Contribution.** Our main contributions are:

(1) We propose a new tuning parameter-free method, unlike existing non-parametric methods for estimating the market noise distribution $F_0$, leveraging the shape constraint that $F_0$ is non-decreasing and assuming only that $F_0$ is $\alpha$-Hölder continuous for some $\alpha \in (0, 1]$.

(2) We derive an upper bound on the total expected regret of order $\widetilde{\mathcal{O}}(T^{\nu(\alpha)} d^{\alpha/2+\alpha})$, where $\nu : (0, 1] \to \mathbb{R}$,

$$\nu(\alpha) \triangleq \frac{2}{2+\alpha} \mathbf{1}\{\alpha \in (0, 1/2)\} + \frac{2\alpha+1}{3\alpha+1} \mathbf{1}\{\alpha \in [1/2, 1]\}, \tag{2}$$

and $\widetilde{\mathcal{O}}$ excludes log factors (under Lipshitzianity of $F_0$, this rates becomes $\widetilde{\mathcal{O}}(d^{1/3}T^{3/4})$), and we provide a thorough assessment of its empirical performance with comparisons to existing algorithms through a number of simulations, as well as an emulation experiment based on real data. Our algorithm shows strong empirical performance: in particular, it dominates the algorithm proposed by Tullii et al. (2024) in their simulation setting up to very large time horizons. Additionally, our algorithm when applied to a real data set obtained by Welltower Inc continues to demonstrate stronger performance than Tullii et al. (2024); Luo et al. (2022) and is competitive with the nonparametric method proposed by Fan et al. (2021), though the latter relies on stronger smoothness assumptions.

(3) Beyond the application of antitonic regression, our work involves establishing a concentration inequality for the uniform norm of the antitonic regression estimator error, which is required to derive the expected regret upper bound. Although the existing literature on isotonic regression explores the rate of convergence of the uniform error, explicit tail probability bounds (stronger than $O_P$ statements) of the type presented in this work (see Theorem 4.8) appear to be missing.

## 1.1 Related Works

The linear valuation model for contextual dynamic pricing, as defined in Equation (1), has been extensively studied under various assumptions. Recent works have explored statistical models—both linear and their

Table 1: Comparison of customer valuation model-based contextual dynamic pricing algorithms with stochastic contexts under the same assumptions on $\theta_0$ and similar smoothness assumptions on $F_0$. Notes: (\$): $\nu(\alpha)$ is defined in Equation (2).

| Method | Regret Upper Bound | Hölder Continuity | Lipschitz Continuity | 2nd Order Smoothness |
|---|---|---|---|---|
| Fan et al. (2021) | $\widetilde{O}((dT)^{\frac{2m+1}{4m-1}})$ | $\times$ | $\times$ | $\checkmark$ |
| Luo et al. (2022) | $\widetilde{O}(T^{3/4}d)$ | $\times$ | $\checkmark$ | $\times$ |
| Tullii et al. (2024) | $\widetilde{O}((dT)^{2/3})$ | $\times$ | $\checkmark$ | $\times$ |
| This Work | $\widetilde{O}(T^{\nu(\alpha)}d^{\alpha/2+\alpha})^{\$}$ | $\checkmark$ | $\checkmark$ | $\times$ |

extensions—for the pricing problem, assuming that $F_0$ (the noise distribution) is either known, partially known[1] (Miao et al., 2019; Ban & Keskin, 2021; Javanmard & Nazerzadeh, 2019; Golrezaei et al., 2019), or fully unknown (Fan et al., 2021; Xu & Wang, 2022; Luo et al., 2022). For comprehensive overviews of dynamic pricing from a broader perspective, we refer readers to Den Boer (2015) and Kumar et al. (2018).

We focus on the results most relevant to our work—specifically, the case where both the parameter $\theta_0$ and the distribution $F_0$ are fully unknown. In this setting, Fan et al. (2021) estimate $F_0$ using kernel methods and derive a regret upper bound of $\widetilde{\mathcal{O}}((dT)^{\frac{2m+1}{4m-1}})$, where $m \geq 2$ denotes the degree of smoothness of $F_0$. In the realm of reinforcement learning, Luo et al. (2022) introduces the Explore-then-UCB strategy, which balances revenue maximization, estimation of the linear valuation parameter, and nonparametric learning of the noise distribution. Under Lipschitz continuity on $F_0$, their approach achieves a regret rate of $\widetilde{\mathcal{O}}(dT^{3/4})$, and under (an additional) second-order smoothness assumption, a regret of $\widetilde{\mathcal{O}}(d^2T^{2/3})$. However, their regret bounds depend on a regularization parameter $\lambda > 0$, which is hard to tune dynamically, and the impact of the choice of $\lambda$ on the regret is not clearly described. Xu & Wang (2022) propose the D2-EXP4 algorithm, which is based on discretizing both the parameter space of $\theta_0$ and $F_0$. With appropriate choices of the discretization parameters, they establish a regret upper bound of $\widetilde{O}(T^{3/4} + \sqrt{d}T^{2/3})$. However, as noted in Xu & Wang (2022, Section 6), they were unable to perform numerical experiments on D2-EXP4 due to the exponential time complexity of the EXP4 learner with a policy set of size $2^{T^{1/4}}$, making their algorithm impractical for application. Furthermore, Assumption 1 of their paper requires their $x_t$ and $\theta_0$ to have *non-negative entries*. While they claim that this assumption entails no loss of generality, this assumption is heavily used in the proofs of Theorem 6 and Theorem 5 of their work, and it is far from clear whether their derivations are generalizable to the situation when such sign constraints are not imposed. While the positivity of covariates can be ensured under boundedness by adding constants and adjusting the intercept parameter, the assumption that all covariates have a positive impact on the valuation is quite unrealistic for any regression model.

In contrast to Fan et al. (2021); Luo et al. (2022); Xu & Wang (2022), we propose a tuning parameter-free policy that achieves a regret upper bound of order $\widetilde{\mathcal{O}}(T^{3/4}d^{1/3})$ when $\alpha = 1$ (i.e. $F_0$ is Lipschitz). Furthermore, estimating the parameters $\theta_0$ and $F_0$ is computationally efficient: $\theta_0$ is estimated using ordinary least squares (OLS), and $F_0$ is estimated via isotonic regression[2] using the Pool Adjacent Violators Algorithm (PAVA) introduced by Robertson et al. (1988), which, in our problem, has a computational complexity of $\mathcal{O}(d^{\alpha/2+\alpha}T^{\nu(\alpha)})$ (see Section 3.1).

Very recent work by Tullii et al. (2024) provides a *UCB-LCB-based* algorithm named VAPE (Valuation Approximation-Price Elimination). The main idea is to update the estimate of $\theta_0$ at time $t$ when $x_t$ is far from previously observed covariate values; otherwise, update the UCB-LCB around $F_0$ and deploy the optimal price. They prove that their regret is upper bounded by $\widetilde{\mathcal{O}}((dT)^{2/3})$ under Lipschitz assumption on $F_0$ (i.e. $\alpha = 1$), which attains the lower bound in $T$, $\Omega(T^{2/3})$ established in Xu & Wang (2022). We summarize the regret upper bounds and the underlying assumptions in Table 1.

---

[1] Meaning $F_0$ is unknown but belongs to a parameterized family.
[2] Alternatively, estimating $S_0 = 1 - F_0$ using antitonic regression.

### 1.2 Notation

For an interval $I = (a, b)$, $a, b \in \mathbb{R}$ we use $|I| = b - a$. For any given matrix $\Sigma \in \mathbb{R}^{d_1 \times d_2}$, we write $\Sigma \succcurlyeq 0$ or $\Sigma \preccurlyeq 0$ if $\Sigma$ or $-\Sigma$ is semi-definite. For any event $A$, we let $\mathbb{I}(A)$ be an indicator random variable which is equal to 1 if $A$ is true and 0 otherwise. For two positive sequences $\{a_n\}_{n \geq 1}, \{b_n\}_{n \geq 1}$, we write $a_n = \mathcal{O}(b_n)$ or $a_n \lesssim b_n$ if there exists a positive constant $C$ such that $a_n \leq C b_n$. We denote $X_n = \mathcal{O}_P(a_n)$ if there exists a $\eta > 0$ and $n_0$ such that $P(X_n \leq a_n) \geq 1 - 1/n^\eta$ for $n \geq n_0$. In addition, we write $a_n = \Omega(b_n)$ or $a_n \gtrsim b_n$ if $a_n/b_n \geq c$ with some constant $c > 0$. Moreover, we let $\widetilde{\mathcal{O}}(\cdot), \widetilde{\Omega}(\cdot)$ represent the same meaning with $\mathcal{O}(\cdot), \Omega(\cdot)$ except for ignoring log factors. For a random variable $x$ we will denote by $f_x$, and $P_x$ its corresponding density function and probability measure, respectively. For a c.d.f. $F$ we will use $S$ to denote $1 - F$. Given a function $h(x, y)$ we write $\mathbb{E}_x[h(x, y)] = \int h(x, y) dP_x(x)$. We say that a function $S$ is $\alpha$-Hölder (continuous) for some constant $\alpha \in (0, 1]$ if $|S(u) - S(v)| \leq L|u - v|^\alpha$ for all $u, v$ in it's domain.

## 2 Problem Setting

We consider the pricing problem where a seller has a single product for sale at each time period $t = 1, 2, \cdots, T$. Here $T$ is the total number of periods (i.e. length of the horizon) and may be unknown to the seller. The *market value of the product* at time $t$ is denoted by $v_t$ and is unknown to the seller. At each period $t$, the seller posts a *price* $p_t \in [p_{\min}, p_{\max}]$ for $0 \leq p_{\min} < p_{\max} < \infty$. If $p_t \leq v_t$, a sale occurs, and the seller collects a revenue of $p_t$. Otherwise, no sale occurs and no revenue is obtained. Let $y_t$ be the response variable that indicates whether a sale has occurred at period $t$:

$$y_t = \mathbf{1}\{v_t \geq p_t\},$$

and let $p_t y_t$ the collected revenue at time $t$. We model the market value $v_t$ as a linear function of the product's observable i.i.d features $x_t \in \mathcal{X} \subset \mathbb{R}^d$

$$v_t = \theta_0^\top x_t + z_t, \tag{3}$$

where $\theta_0 \in \mathbb{R}^d$ is an unknown parameter (which includes the intercept term), and $z_t$ are i.i.d sequence of idiosyncratic noise drawn from an unknown distribution $F_0$ with mean 0 and bounded support

$$\mathcal{U} \triangleq (\inf\{z \in \mathbb{R} : F_0(z) > 0\}, \sup\{z \in \mathbb{R} : F_0(z) < 1\}). \tag{4}$$

We assume that the first entry in $x_t$ equals 1 to account for the intercept term in $\theta_0$. The overall procedure is summarized in Box 1. The expected revenue for any offered price $p$ given $x_t$ is

$$r_t(p) \triangleq \mathbb{E}_{z_t}(p\mathbf{1}\{v_t > p\} \mid x_t) = pP_{z_t}(v_t > p \mid x_t) = pS_0(p - \theta_0^\top x_t), \quad S_0 = 1 - F_0.$$

Note that, since $S_0$ is a survival function, it is non-increasing. The optimal price $p_t^*$ at time $t$ is defined by a maximizer of the expected revenue function at the round,

$$p_t^* \in \operatorname{argmax}_{p \in [p_{\min}, p_{\max}]} pS_0\left(p - \theta_0^\top x_t\right). \tag{5}$$

Note that $p_t^* = p_t^*(x_t)$, depends on $x_t$. The regret at step $t$ is defined by the difference between the expected revenues from the optimal price $p_t^*$ and the offered price $p_t$: $r_t(p_t^*) - r_t(p_t)$. In other words, we consider the problem of maximizing revenue as minimizing the following maximum regret

$$R(T) \triangleq \mathbb{E}\left[\sum_{t=1}^T p_t^* \mathbf{1}\{p_t^* \leq v_t\} - p_t \mathbf{1}\{p_t \leq v_t\}\right],$$

where the expectation is taken with respect to the idiosyncratic noise $z_t$, the covariates $x_t$, and the offered prices $p_t$ that depend on the specific policy.

---

**Box 1: Contextual Pricing Dynamic**

For each sales round $t = 1, \ldots, T$:

(1) The seller observes a context vector $x_t \in \mathbb{R}^d$.

(2) The seller offers a price $p_t$ based on $x_t$ and the previous sales records $\{(x_\tau, p_\tau, y_\tau)\}_{\tau=1}^{t-1}$.

(3) Simultaneously, the customer evaluates the product at $v_t$, which is not known to the seller.

(4) The seller observes $y_t = \mathbf{1}\{v_t \geq p_t\}$, indicating whether the product was sold.

---

As the firm's goal is to design a policy that sets prices $p_t$ as close as possible to the optimal prices $p_t^*$ defined in Equation (5), we first estimate $(\theta_0, S_0)$ and then we plug in the estimate as in Equation (6) to get an estimated optimal price $p_t$. Accurate estimation of $(\theta_0, S_0)$ thereby ensures that the resulting policy incurs low regret.

## 3 Proposed Algorithm

We employ an epoch-based design (also known as the doubling trick) that segments the given horizon $T$ into several clusters of rounds, called *epochs* or *episodes*, and executes identical pricing policies on a per-epoch basis. Let $\mathcal{J}_1 = \{0, 1, \ldots, \tau_1 - 1\}$ be the first episode, where $\tau_1$ is a prefixed constant. For $k = 2, \ldots, K = \lceil \log(T/\tau_1) + 1 \rceil$, define $\tau_k = \tau_1 2^{k-1}$, and $\mathcal{J}_k = \{\tau_k - \tau_1, \ldots, \tau_{k+1} - \tau_1 - 1\}$ the set of times in the $k$-th episode.

We partition $\mathcal{J}_k$ into two sub-phases, $\mathcal{J}_k = E_k \cup E_k'$, where $E_k$ represents the *exploration phase*, dedicated to collecting data for estimating the parameters $(\theta_0, S_0)$, while $E_k'$ denotes the *exploitation phase*, during which we apply the optimal prices based on the estimated parameters $(\widehat{\theta}_k, \widehat{S}_k)$. The length of the exploration phase, $|E_k|$, is set to $\lceil d^{\alpha/2 + \alpha} \tau_k^{\nu(\alpha)} \rceil$, chosen to minimize the expected regret $R(T)$. Specifically, as we show in the proof of Theorem 4.10, if $|E_k| = d^\xi \tau_k^\eta$ for some $\xi, \eta \in (0, 1)$, then $R(T)$ is minimized if $\xi$ and $\eta$ satisfy the condition $d^\xi \tau_k^\eta = \lceil d^{\alpha/2 + \alpha} \tau_k^{\nu(\alpha)} \rceil$. $E_k$ is further divided into two equal-sized intervals $I_k$ and $\widetilde{I}_k$. In $I_k$ we collect data to estimate $\theta_0$. In $\widetilde{I}_k$ we collect data to estimate $S_0$. The details are stated in Algorithm 1, and a picture of a general episode $\mathcal{J}_k$ is shown in Figure 1. In the following portion of this section, we examine the details of exploration-exploitation for a fixed episode $k$.

**Estimation of $\theta_0$.** For all $t \in I_k$ the seller observe $x_t$, deploy $p_t \sim \text{unif}(p_{\min}, p_{\max})$ and observes $y_t = \mathbf{1}\{p_t \leq v_t\}$. Let $H = p_{\max} - p_{\min}$ and estimate

$$\widehat{\theta}_k = \text{OLS}\{(x_t, Hy_t)\}_{t \in I_k} = \arg\min_\theta \frac{1}{|I_k|} \sum_{t \in I_k} (Hy_t - \theta^\top x_t)^2.$$

**Estimation of $S_0$.** For $t \in \widetilde{I}_k$, the seller observe $x_t$, sample $w_t \sim \text{unif}(\mathcal{U})$, propose a price $p_t = w_t + \widehat{\theta}_k^\top x_t$ and observes $y_t = \mathbf{1}\{p_t \leq v_t\}$. Estimate

$$\widehat{S}_k = \text{Antitonic}\{(w_t, y_t)\}_{t \in \widetilde{I}_k} = \arg\min_{S \in \mathcal{S}} \sum_{t \in \widetilde{I}_k} (y_t - S(w_t))^2,$$

where $\mathcal{S}$ is the set of non-increasing function in $\mathbb{R}$.

**Exploitation.** For every $t \in E_k'$, observe $x_t$, set

$$p_t \in \text{argmax}_{p \in [p_{\min}, p_{\max}]} \, p\widehat{S}_k(p - \widehat{\theta}_k^\top x_t), \tag{6}$$

and get reward $p_t \mathbf{1}\{p_t \leq v_t\}$.

### 3.1 Complexity of the antitonic regression

The algorithmic complexity for estimating $S_0$ is $\mathcal{O}(d^{\alpha/2 + \alpha} T^{\nu(\alpha)})$. Indeed by Grotzinger & Witzgall (1984); Tibshirani et al. (2011) the computational complexity for the antitonic estimator is $\mathcal{O}(n)$, where $n$ is the sample size. In our case, the estimation of $S_0$ happens in (half of) the exploration phase which has length proportional to $d^{\alpha/2 + \alpha} \tau_K^{\nu(\alpha)} = d^{\alpha/2 + \alpha} 2^{K\nu(\alpha)}$, then, using that $K \propto \log_2(T)$, we have $n \propto d^{\alpha/2 + \alpha} T^{\nu(\alpha)}$.

---

**Algorithm 1** Semi-Parametric Pricing

---

**Input:** Time horizon $T$ and length of the first epoch, $\tau_1$; the Hölder exponent $\alpha$ of $S_0$ and the corresponding $\nu(\alpha)$ defined in Equation (2); the minimum and maximum of price search range, $p_{\min}$ and $p_{\max}$, $H = p_{\max} - p_{\min}$; $\mathcal{U}$ defined in Equation (4).

Set $K = \lceil \log{(T/\tau_1)} + 1 \rceil$.

**for** epoch $k = 1, 2, \ldots, K$ **do**

  $\tau_k \leftarrow \tau_1 2^{k-1}$, length of episode $k$

  $a_k \leftarrow \lceil d^{\alpha/2+\alpha} \tau_k^{\nu(\alpha)}/2 \rceil$, length of exploration phase

  $I_k \leftarrow \{\tau_k - \tau_1, \ldots, \tau_k - \tau_1 + a_k - 1\}$

  $\widetilde{I}_k \leftarrow \{\tau_k - \tau_1 + a_k, \ldots, \tau_k - \tau_1 + 2a_k - 1\}$

  $E_k \leftarrow I_k \cup \widetilde{I}_k$ indexes of the exploration phase

  $E'_k \leftarrow \{\tau_k - \tau_1 + 2a_k, \ldots, \tau_{k+1} - \tau_1 - 1\}$ indexes of the exploitation phase.

  **for** $t \in I_k$ **do**

    Observe $x_t$

    Set $p_t \sim \text{unif}(p_{\min}, p_{\max})$.

    Get $y_t \leftarrow \mathbf{1}\{p_t \leq v_t\}$

  **end for**

  $\widehat{\theta}_k \leftarrow \text{OLS}\{(x_t, Hy_t)\}_{t \in I_k}$

  **for** $t \in \widetilde{I}_k$ **do**

    Observe $x_t$

    Sample $w_t \sim \text{unif}(\mathcal{U})$

    Set $p_t \leftarrow w_t + \widehat{\theta}_k x_t$

    Get $y_t \leftarrow \mathbf{1}\{p_t \leq v_t\}$

  **end for**

  $\widehat{S}_k \leftarrow \text{Antitonic}\{(w_t, y_t)\}_{t \in \widetilde{I}_k}$

  **for** $t \in E'_k$ **do**

    Observe $x_t$

    Set price $p_t$ as defined in Equation (6).

    Get $y_t \leftarrow \mathbf{1}\{p_t \leq v_t\}$

  **end for**

**end for**

---

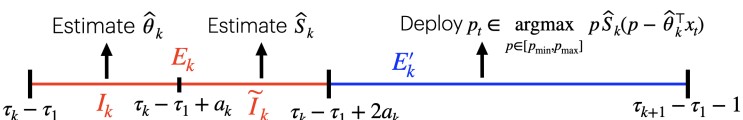

Figure 1: Picture of a general episode $\mathcal{J}_k$, $k = 1, 2, \ldots, K$.

# 4 Regret Analysis

Before proceeding with the regret analysis we need to discuss the convergence rates of $\widehat{\theta}_k$ to $\theta_0$ and $\widehat{S}_k$ to $S_0$. We present our main theorems and proofs. We defer to the Appendix for the missing proofs.

## 4.1 Estimation of $\theta_0$

**Assumption 4.1** (Bounded parameter space)**.** The parameter $\theta_0 \in \mathbb{R}^d$ is an interior point of $\Theta$ and the parameter space $\Theta$ is a compact convex set.

**Assumption 4.2** (Bounded i.i.d. contexts). (a) $x_t \in \mathcal{X} \subset \mathbb{R}^d$ is i.i.d. drawn a distribution that does not involve $\theta_0$ and $F_0$, and for all $x_t \in \mathcal{X}$, $\|x_t\|_2 \leq R_{\mathcal{X}}$ for some unknown $R_{\mathcal{X}} > 0$. (b) There exists reals $c_{\min}, c_{\max} > 0$, s.t. $c_{\min}\mathbb{I}_d \preccurlyeq \Sigma \preccurlyeq c_{\max}\mathbb{I}_d$, where $\Sigma \triangleq \mathbb{E}x_t x_t^\top$ and $\mathbb{I}_d$ the $d \times d$ identity matrix.

Assumptions 4.1 and 4.2 are standard in the dynamic pricing literature Javanmard & Nazerzadeh (2019); Xu & Wang (2022); Luo et al. (2022); Fan et al. (2021). More precisely, the i.i.d. assumption of the context and the prices $\{p_t\}_{t \in I_k}$, is used in Lemma 4.3 to derive a concentration inequality for the estimation of $\theta_0$.

**Lemma 4.3.** *[Fan et al. (2021, Lemma 4.1)] Let $n_k \triangleq |I_k|$ for simplicity of notation. Under Assumptions 4.1 and 4.2, there exist $c_0, c_1 > 0$ depending only on absolute constants given in assumptions, such that for any episode $k$, as long as $n_k \geq c_0 d$, with probability at least $1 - Q_{n_k}$, with $Q_{n_k} \triangleq 2e^{-c_1 c_{\min}^2 n_k / 16} + \frac{2}{n_k}$, it holds that*

$$\|\widehat{\theta}_k - \theta_0\|_2 \leq C_{\theta_0}\sqrt{d\log n_k / n_k} \triangleq R_{n_k},$$

*where $C_{\theta_0} \triangleq \frac{8\max\{R_{\mathcal{X}}, 1\}\left(R_{\mathcal{X}}R_{\theta_0} + p_{\max} - p_{\min}\right)}{c_{\min}}$.*

### 4.2 Estimation of $S_0$ via antitonic regression

In this section, we provide a uniform convergence result of $\widehat{S}_k$ to $S_0$. For simplicity of notation, we re-index $\widetilde{I}_k$ as $\mathcal{T} = \{1, 2, \ldots, n\}$, and we denote $\widehat{\theta}_k$ by $\theta$, which was estimated using data $\{(p_t, x_t, y_t)\}_{t \in I_k}$ independent of $\mathcal{T}$. In this section, all the results must be considered as conditioned on $\theta = \widehat{\theta}_k$. We report in Box 2 a more detailed explanation of the estimation of $S_0$ during the exploration phase in $\mathcal{T}$ as expounded in the box highlighted in Algorithm 1.

---

**Box 2: Sample collection for estimating $S_\theta$**

For each $t \in \mathcal{T}$ do:

(1) Observe $x_t$.

(2) The customer samples $z_t$ and evaluate $v_t = \theta_0^\top x_t + z_t$, unknown to the firm.

(3) The firm samples $w_t \sim \text{unif}(\mathcal{U})$ independent of everything else and defines $p_t = w_t + \theta^\top x_t$.

(4) Observe $y_t = \mathbf{1}\{p_t \leq v_t\}$.

---

This produces a set of data points $\{(p_t, x_t, y_t)\}_{t \in \mathcal{T}}$ that we are going to use for estimating $S_0$. To estimate $S_0$ we would need to know $\theta_0$ in advance, indeed remember that $\mathbb{E}(y_t \mid p_t, x_t) = S_0(p_t - \theta_0^\top x_t)$, which design points $\{p_t - \theta_0^\top x_t\}_{t \in \mathcal{T}}$ depend on $\theta_0$. However, our knowledge is limited to an approximation $\theta$ of $\theta_0$, and the observable design points are $p_t - \theta^\top x_t = w_t$. This implies that we are only able to estimate $S_\theta(u) \triangleq \mathbb{E}(y_t \mid w_t = u)$. We then estimate $S_\theta(\cdot)$ considering $y_t$ as coming from a sample in the ordinary current status model, where the data has the form $(w_t, y_t) \overset{i.i.d}{\sim} (w, y)$, the observation times have uniform density $f_w$ and where $y_t = 1$ with probability $S_\theta(w_t)$ at observation $w_t$.

*Remark* 4.4 (**The choice of the design points $w_t$**). The choice of the distribution for $w_t$ is motivated by the fact that when the density of design points is uniform, we obtain convergence guarantees for the estimator of $S_0$. Specifically, as mentioned by Mösching & Dümbgen (2020), if the density of the design points is bounded away from zero – which holds for the uniform distribution – then $\{w_t\}_{t \in \mathcal{T}}$ are "asymptotically dense" within any interval contained in $\mathcal{U}$ (defined in Equation (4)). Ensuring that the design points have a density bounded away from zero is a sufficient condition for the convergence result in Theorem 4.8 (see Lemma A.2 for further details). However, this choice is not restrictive; any distribution whose density is bounded away from zero in $\mathcal{U}$ would still satisfy the convergence result. The only consequence of using a non-uniform density is that the regret $R(T)$ will depend on the multiplicative constant $C_2 = \inf_{u \in \mathcal{U}} f_w(u)$, which may be different from $1/|\mathcal{U}|$ if $f_w$ is not the uniform density in $\mathcal{U}$. Furthermore, since each episode $\mathcal{J}_k$ is independent of any other episode $\mathcal{J}_{k'}$ for $k \neq k'$, it is possible to select a different density $f_w^{(k)}$ for each episode $k$, provided that it remains bounded away from zero in $\mathcal{U}$. An interesting extension of our work would be to adaptively update the design density $f_w^{(k)}$ based on the previous design $f_w^{(k-1)}$ in an *optimal* manner, namely that $f_w^{(k)}$ converges to the optimal design density as $k \to \infty$, that is the density that minimizes the integrated mean square error. This approach, known as *sequential optimal design*, has been extensively studied in the literature (see, e.g.,

Müller (1984); Zhao & Yao (2012); Bracale et al. (2024)). A key advantage of an optimal design algorithm is that it dynamically allocates more data to regions where the estimation of $S_0$ is less accurate, thereby progressively improving its precision. However, it is important to note that while this adaptive approach can optimize the multiplicative constant in the regret bound, it does not affect the rate of the regret itself.

*Remark* 4.5 (**The difference between the conditional distributions of $y|(p,x)$ and $y|w$**)*. We want to highlight the difference between the conditional distributions of $y|(p,x)$ and $y|w$. The first is independent of $\theta$, indeed we have that*

$$\mathbb{E}_z[y|p,x] = \mathbb{E}[\mathbf{1}\{p \le \theta_0^\top x + z\}|p,x] = S_0(p - \theta_0^\top x),$$

while, the distribution of $y|w$ depends on $\theta$ because $y = \mathbf{1}\{p \le v\}$ where $p = w + \theta^\top x$, and, since data $(w,y)$ is generated as in Box 2, we have that

$$
\begin{aligned}
S_\theta(u) = \mathbb{E}_z(y \mid w = u) &= \mathbb{E}_{(p,x)}(\mathbb{E}_z(y \mid p,x) \mid w = u) \\
&= \mathbb{E}_{(p,x)}(S_0(p - \theta_0^\top x) \mid w = u) \\
&= \mathbb{E}_{(p,x)}(S_0(p - \theta^\top x + \theta^\top x - \theta_0^\top x) \mid w = u) \\
&= \mathbb{E}_x(S_0(u + (\theta - \theta_0)^\top x) \mid w = u) \\
&= \int S_0(u + (\theta - \theta_0)^\top x) dP_{x|w=u}(x) \\
&= \int S_0(u + (\theta - \theta_0)^\top x) dP_x(x),
\end{aligned}
\tag{7}
$$

where the first equality is by definition, in the second we use the tower property and in the last equality, we use that $w_t$ is sampled independently of $x_t$. Note from Equation (7) that $S_\theta(\cdot)$ is non-increasing for all $\theta$ because $S_0 = 1 - F_0$, being a survival function, is non-increasing.

**Proposition 4.6.** *$S_\theta$ is non-increasing for every $\theta \in \mathbb{R}^d$. Moreover, if $S_0$ is $\alpha$-Hölder with $\alpha \in (0,1]$, then $S_\theta$ is $\alpha$-Hölder uniformly in $\theta \in \mathbb{R}^d$ and $|S_\theta(u) - S_0(u)| \lesssim \|\theta - \theta_0\|_2^\alpha$ uniformly in $u \in \mathbb{R}$.*

Proposition 4.6 is crucial because it tells us that we can estimate $S_\theta$ under the antitonic constraint and that $S_\theta$ is close to $S_0$ as long as $\theta$ is close to $\theta_0$, which will be used to prove Theorem 4.10. Guided by Proposition 4.6, we estimate $S_\theta$ using antitonic regression, denoted as

$$\widehat{S}_\theta \triangleq \operatorname{argmin}_{S \in \mathcal{S}} \sum_{t \in \mathcal{T}} (y_t - S(w_t))^2, \tag{8}$$

where $\mathcal{S}$ is the set of non-increasing functions in $\mathbb{R}$. The minimizer $\widehat{S}_\theta$ is a piecewise constant function with jumps at a subset of $\{w_t : t \in \mathcal{T}\}$. The order statistics on which $\widehat{S}_\theta$ is based are the order statistics of the values $w_t$ and the values of the corresponding $y_t$. To be more specific, let $u_1 < u_2 < \cdots < u_m$ the different value of the observed $\{w_t\}_{t \in \mathcal{T}}$. For $j = 1, \ldots, m$ set

$$o_j = \#\{t : w_t = u_j\}, \quad \widehat{y}_j = \frac{1}{o_j} \sum_{i:w_i = u_j} y_i.$$

For every $1 \le r \le s \le m$ let

$$o_{rs} \triangleq \sum_{j=r}^s o_j = \# \{t : u_r \le w_t \le u_s\}, \quad \widehat{y}_{rs} = \frac{1}{o_{rs}} \sum_{j=r}^s o_j \widehat{y}_j.$$

It is well known that $\widehat{S}_\theta = (\widehat{S}_\theta(u_1), \widehat{S}_\theta(u_2), \ldots, \widehat{S}_\theta(u_m))$ may be represented by the following minimax and maximin formulae, see Robertson et al. (1988): for $1 \le j \le m$

$$\widehat{S}_\theta(u_j) = \min_{r \le j} \max_{s \ge j} \widehat{y}_{rs} = \max_{s \ge j} \min_{r \le j} \widehat{y}_{rs}.$$

The $\widehat{S}_\theta$ is also known as the antitonic regression on data $\{(w_t, y_t)\}_{t \in \mathcal{T}}$, and we will denote it as

$$\widehat{S}_\theta = \operatorname{Antitonic}\{(w_t, y_t)\}_{t \in \mathcal{T}}.$$

We are now prepared to demonstrate the convergence of $\widehat{S}_k$ to $S_0$. To establish this result, we require that $S_\theta$ is $\alpha$-Hölder for some $\alpha \in (0,1]$ uniformly in $\theta$. According to Proposition 4.6, this condition is satisfied provided we make the following assumption:

**Assumption 4.7.** $|S_0(u) - S_0(v)| \leq C_1|u-v|^\alpha$ for some $\alpha \in (0,1]$, $C_1 > 0$ and for all $u, v \in \mathbb{R}$.

**Theorem 4.8.** *Let $\{(w_t, y_t)\}_{t\in\mathcal{T}}$ be as defined in Box 2 and let Assumption 4.7 hold. Then for every $\kappa > 0$ and $\gamma > 2$ there exists $n_0 = n_0(\gamma, \kappa, \alpha) \in \mathbb{N}$ and $C = C(C_1, |\mathcal{U}|, \kappa, \alpha) > 0$ (where $\mathcal{U}$ is defined in Equation (4)) such that*

$$\mathbb{P}\left\{\sup_{u\in\mathcal{U}_n} |\widehat{S}_\theta(u) - S_\theta(u)| \leq C\rho_n^{\alpha/(2\alpha+1)}\right\} \geq 1 - \frac{1}{n^{\gamma-2}}, \quad n \geq n_0,$$

*where $\rho_n = \log(n)/n$ and $\mathcal{U}_n = \{u \in \mathcal{U} : [u \pm \delta_n] \subset \mathcal{U}\}$, with $\delta_n = \kappa\rho_n^{1/(2\alpha+1)}$.*

*Remark* 4.9. Our Theorem 4.8 parallels Theorem 3.3 in Mösching & Dümbgen (2020), with the key distinction being the nature of the observed response variable. While Mösching & Dümbgen (2020) directly observes the response variable (which corresponds to our valuation $v_t$), we observe the binary indicator $y_t = \mathbf{1}\{p_t \leq v_t\}$. This difference simplifies our proof, as it only requires establishing a concentration inequality for $|\widehat{y}_{rs} - \bar{S}_{rs}(\theta)|$, where

$$\bar{S}_{rs}(\theta) \triangleq \frac{1}{o_{rs}} \sum_{j=r}^s o_j S_\theta(w_j).$$

In our setting, this concentration inequality can be readily obtained using Hoeffding's inequality uniformly over $\theta$. Specifically, as demonstrated in Lemma A.1, for any constant $D > 1$, $\mathbb{P}\{M_n(\theta) \leq (D\log n)^{1/2}\}$ is at least $1 - (n+1/n^D)^2$, where $M_n(\theta) \triangleq \max_{1\leq r\leq s\leq m} o_{rs}^{1/2}|\widehat{y}_{rs} - \bar{S}_{rs}(\theta)|$.

### 4.3 Regret Upper Bound

We are now ready to establish an upper bound on the expected regret for our Algorithm 1.

**Theorem 4.10.** *Suppose that Assumptions 4.1, 4.2 and 4.7. For sufficiently large $T$ the cumulative regret of Algorithm 1, $R(T)$ has upper bound of order*

$$\begin{cases} \mathcal{O}(T^{\frac{2}{2+\alpha}} d^{\frac{\alpha}{\alpha+2}} \log^{\frac{\alpha}{2\alpha+1}}(dT)), & \alpha \in (0, 1/2), \\ \mathcal{O}(T^{\frac{2\alpha+1}{3\alpha+1}} d^{\frac{\alpha}{\alpha+2}} \log^{\frac{\alpha}{2}}(dT)), & \alpha \in [1/2, 1]. \end{cases}$$

*Proof.* **Sketch.** Fix $k \geq 2$ and define $n_k = |I_k|$ and $\widetilde{n}_k = |\widetilde{I}_k|$ and $a_k = |E_k| = n_k + \widetilde{n}_k$. Let $S_0(p \mid x) \triangleq S_0(p - \theta_0^\top x)$ and $\widehat{S}_k(p \mid x) \triangleq \widehat{S}_k(p - \widehat{\theta}_k^\top x)$. For the exploration phase $\mathbb{E}[\sum_{t\in E_k} r_t(p_t^*) - r_t(p_t)] \leq p_{\max}|E_k| \lesssim |E_k|$. Now fix $t \in E_k'$

$$r_t(p_t^*) - r_t(p_t) = p_t^* S_0(p_t^* \mid x_t) - p_t S_0(p_t \mid x_t)$$
$$= \left\{p_t^* S_0(p_t^* \mid x_t) - p_t^*\widehat{S}_k(p_t^* \mid x_t)\right\} + \underbrace{\left\{p_t^*\widehat{S}_k(p_t^* \mid x_t) - p_t\widehat{S}_k(p_t \mid x_t)\right\}}_{\leq 0 \text{ by Equation (6)}} + \left\{p_t\widehat{S}_k(p_t \mid x_t) - p_t S_0(p_t \mid x_t)\right\}$$
$$\leq p_{\max}\left|S_0(p_t^* \mid x_t) - \widehat{S}_k(p_t^* \mid x_t)\right| + p_{\max}\left|\widehat{S}_k(p_t \mid x_t) - S_0(p_t \mid x_t)\right| = R_{k,t}(p_t^*) + R_{k,t}(p_t),$$

where $R_{k,t}(q) \triangleq |\widehat{S}_k(q - \widehat{\theta}_k^\top x_t) - S_0(q - \theta_0^\top x_t)|$ for $q \in \{p_t^*, p_t\}$, $t \in E_k'$.

**Lemma 4.11.** *If Assumptions in Theorem 4.10 hold, for $k$ sufficiently large we have $\mathbb{E}(R_{k,t}(q)) \lesssim \left(\log\widetilde{n}_k/\widetilde{n}_k\right)^{\alpha/2\alpha+1} + (d\log n_k/n_k)^{\alpha/2}$ for $q \in \{p_t^*, p_t\}$ with $t \in E_k'$.*

Let $k \geq k_0$ for $k_0$ be sufficiently large as in Lemma 4.11. Summing up for all $t \in E_k'$, and merging with the exploration phase of episode $k$ we get

$$\mathbb{E}\left[\sum_{t\in\mathcal{J}_k} r_t(p_t^*) - r_t(p_t)\right] \lesssim |E_k| + |E_k'|\left[\left(\log\widetilde{n}_k/\widetilde{n}_k\right)^{\alpha/2\alpha+1} + (d\log n_k/n_k)^{\alpha/2}\right].$$

Using that $n_k = \widetilde{n}_k = \frac{1}{2}a_k = \frac{1}{2}|E_k| = \frac{1}{2}d^\xi(\tau_1 2^{k-1})^\nu \propto d^\xi 2^{k\nu}$ for $\xi, \nu > 0$ to be determined such that they minimize the total regret, and that $|E_k'| \leq |\mathcal{J}_k| = \tau_1 2^{k-1} \propto 2^k$ we get that the RHS of the last inequality is bounded above by

$$d^\xi 2^{k\nu} + d^{-\frac{\xi\alpha}{2\alpha+1}} 2^{k(1-\frac{\nu\alpha}{2\alpha+1})}[k + \log(d)]^{\frac{\alpha}{2\alpha+1}} + d^{\frac{\alpha}{2}(1-\xi)} 2^{k(1-\frac{\nu\alpha}{2})}[k + \log(d)]^{\frac{\alpha}{2}}.$$

Here we prove the result for $\alpha > 1$. We defer to Appendix A.3 for the complete proof. Consider the exponents of the factor $d$, i.e. $\xi, -\frac{\xi\alpha}{2\alpha+1}$ and $\frac{\alpha}{2}(1-\xi)$. As the second exponent is always negative, the regret is minimized with respect to $d$ if we equalize the first and the third exponent, i.e. $\xi = \frac{\alpha}{2}(1-\xi)$ to get $\xi^* = \frac{\alpha}{\alpha+2}$. Similarly for the exponents of the exponential factor $2^k$, i.e. $\nu, 1-\frac{\nu\alpha}{2\alpha+1}$ and $1-\frac{\nu\alpha}{2}$, we equalize the first two factors, to get $\nu^* = \frac{2\alpha+1}{3\alpha+1}$. Consequently, the expected regret in episode $k$, $\mathbb{E}\left[\sum_{t\in\mathcal{J}_k} r_t(p_t^*) - r_t(p_t)\right]$ is upper bounded by

$$2^{k\frac{2\alpha+1}{3\alpha+1}}(d^{\frac{\alpha}{\alpha+2}} + d^{-\frac{\alpha^2}{(2\alpha+1)(\alpha+2)}}[k+\log(d)]^{\frac{\alpha}{2\alpha+1}} + d^{\frac{\alpha}{\alpha+2}}[k+\log(d)]^{\frac{\alpha}{2}}) \lesssim 2^{k\frac{2\alpha+1}{3\alpha+1}} d^{\frac{\alpha}{\alpha+2}}[k+\log(d)]^{\frac{\alpha}{2}},$$

where we used that $\frac{\alpha}{2\alpha+1} < \frac{\alpha}{2}$ for $\alpha \in (1/2, 1]$. Putting together the phases and using that $K = \lceil \log(T/\tau_1) + 1 \rceil$ we get

$$R(T) = \mathbb{E}\left[\sum_{k=k_0}^{K}\sum_{t\in\mathcal{J}_k} r_t(p_t^*) - r_t(p_t)\right] \lesssim 2^{K\frac{2\alpha+1}{3\alpha+1}} d^{\frac{\alpha}{\alpha+2}}[K+\log(d)]^{\frac{\alpha}{2}} \lesssim T^{\frac{2\alpha+1}{3\alpha+1}} d^{\frac{\alpha}{\alpha+2}} \log^{\frac{\alpha}{2}}(dT).$$

$\square$

## 5 Simulations

We first perform simulations for theoretical validation in Section 5.1 and a simulation to compare our algorithm with the minimax algorithm by Tullii et al. (2024) and Fan et al. (2021) algorithm in Section 5.2.

### 5.1 Simulation for theoretical validation

To this end, we replicate the simulation settings used by Fan et al. (2021). We set $\mathcal{U} = (-1/2, 1/2)$ (known), the feature dimension $d = 3$ (known), the distribution of $X_t \sim \text{Unif}(-\sqrt{2/3}, \sqrt{2/3})^{\times d}$ (unknown), and the coefficient $\theta_0^\top = (\alpha_0, \beta_0^\top)$ (unknown), where $\alpha_0 = 3, \beta_0 = (2/3, 2/3, 2/3)$. We also choose $p_{\min} = 0$ and $p_{\max} = 5$ (known). For $F_0 : \mathcal{U} \to \mathbb{R}$ we consider different choices:

(1) $\boldsymbol{\alpha < 1}$: $F_{0,\alpha}(z) = 1/2 + (1/2)^{1-\alpha} \text{sign}(z)|z|^\alpha$ for $z \in \mathcal{U}$, for $z > 1/2$, for $\alpha \in \{1/3, 1/2, 3/4\}$.

(2) $\boldsymbol{\alpha = 1}$: we use four choices of $F_0$: a Gaussian $N(0,1)$ truncated at $\mathcal{U}$, the c.d.f. used by Fan et al. (2021) with density $f_0(z) = 6\left(\frac{1}{4} - z^2\right)\mathbf{1}\{z \in \mathcal{U}\}$, a Laplace with location 0 and scale 0.2 truncated at $\mathcal{U}$, and a Cauchy with location 0 and scale 0.2 truncated at $\mathcal{U}$.

We start with $\tau_1 = 100$ and compute $K = 8$ total episodes. At every time $t$ we follow Algorithm 1 to compute $p_t$, with the additional computation of the oracle $p_t^*$ and the corresponding cumulative regret $\text{Reg}(t) = \sum_{j=1}^{t} p_t^* S_0(p_t^* - \theta_0^\top x_t) - p_t S_0(p_t - \theta_0^\top x_t)$. We repeated the experiment 36 times and we computed the mean and the 95% confidence interval in a $\log_2 - \log_2$ plot. As illustrated in Figure 2, we validate our approach by comparing the estimated slope of the linear regression of $\log_2(t)$ versus $\log_2(\text{Reg}(t))$ with the theoretical upper bound rate. Due to space constraints, the plot corresponding to the $F_0$ used by Fan et al. (2021) with density $f_0(z) = 6\left(\frac{1}{4} - z^2\right)\mathbf{1}\{z \in \mathcal{U}\}$ is provided in Appendix B.

*Remark* 5.1 (Dependence on $\tau_1$). Although $\tau_1$ is generally considered a non-tuning parameter – typically set to 1 in dynamic pricing algorithms that use the doubling trick (see, e.g., Javanmard & Nazerzadeh (2019); Javanmard et al. (2020; 2024)) – we have included an additional simulation in Figure 3 to demonstrate the robustness of our algorithm with respect to this parameter.

### 5.2 Comparison with Tullii et al. (2024) under Lipschitz assumption of $F_0$

We first recall that the regret upper bound by Tullii et al. (2024) is of order $\widetilde{\mathcal{O}}(T^{2/3})$ under Lipschitz assumption on $F_0$ ($\alpha = 1$), which is smaller than our regret upper-bound $\widetilde{\mathcal{O}}(T^{3/4})$. For this reason, we perform the following simulations.

In their work, Tullii et al. (2024, Supplemenary Material A) compared their VAPE method to the kernel-based method by Fan et al. (2021) that is: they built a dataset of 5 contexts belonging to $\mathbb{R}^3$ generated by a

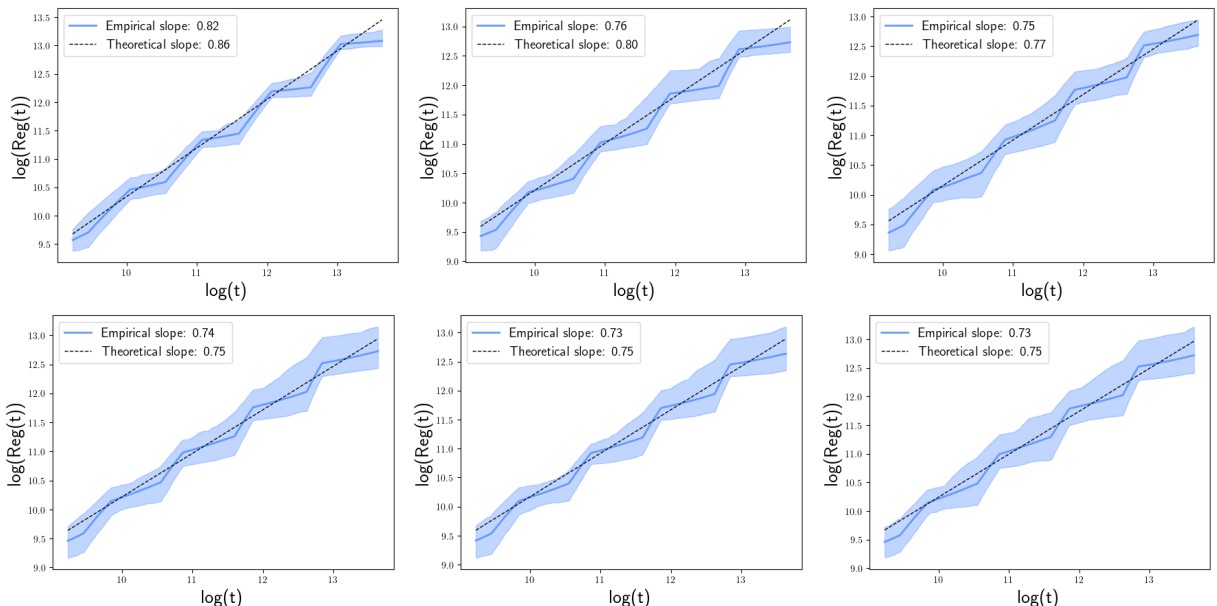

Figure 2: This plot shows the total expected regret (blue line) with $F_{0,\alpha}$, for $\alpha \in \{1/3, 1/2, 3/4\}$ in the first row, and a Gaussian, Laplace, and Cauchy c.d.f. in the second row (from the left to the right). We repeated the simulation 36 times and displayed the corresponding 95% confidence intervals. The plot is in $\log_2$ - $\log_2$ scale to show the regret rate (empirical slope): a slope of $\eta$ indicated an $\mathcal{O}(T^\eta)$ regret. The black dashed line corresponds to our theoretical regret upper bound.

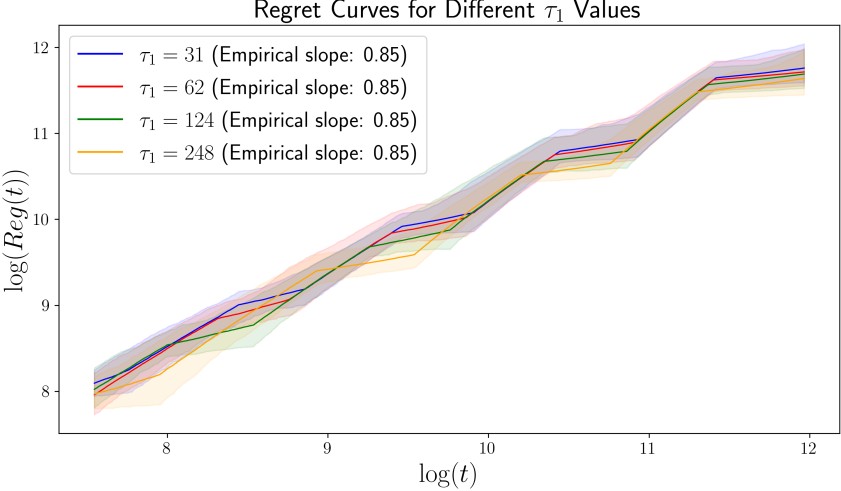

Figure 3: This plot in $\log_2 - \log_2$ scale shows the cumulative regret over time up to $T = 4000$ for different values of $\tau_1 \in \{31, 62, 124, 248\}$ and with $F_{0,1/3}$ for which theoretical regret rate is 0.86. For each value of $\tau_1$, we repeated the simulation 36 times and displayed the corresponding 95% confidence intervals. As we can see, the regret remains similar across different values of $\tau_1$ and the the empirical slopes are close to the theoretical regret rate.

canonical Gaussian distribution and subsequently normalized. Throughout the run, the contexts are chosen from this set uniformly at random, while the noise term is picked from a Gaussian distribution truncated between $-1$ and $1$ with mean 0 and variance 0.1. Similarly, also the parameter $\theta_0$ is a normalized vector initially drawn from a Gaussian distribution. Note that for this simulation, the error distribution is twice differentiable (i.e. smoother than what Tullii et al. (2024) and us allow in our theory), then the kernel-based

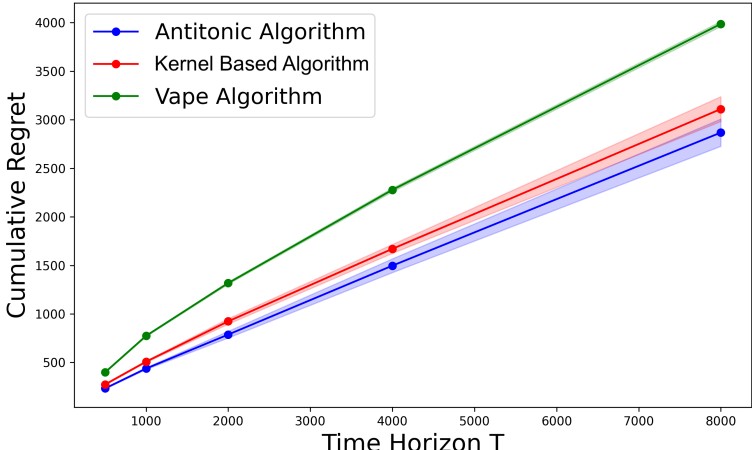

Figure 4: Regret comparison in the simulation setting of Tullii et al. (2024).

method by Fan et al. (2021) is applicable with smoothness parameter $m = 2$. Tullii et al. (2024) showed that the kernel-based method has stronger performance.

We apply our antitonic regression-based algorithm with $\alpha = 1$, using the same code and simulation setting provided by Tullii et al. (2024, Supplemenary Material A). The algorithm has been tested on time horizons $T \in [500, 1000, 2000, 4000, 8000]$. We computed the regret 36 times and the corresponding 95% confidence interval. In Figure 4 we show the results. Although the kernel-based method by Fan et al. (2021) applies to distributions of the error that are at least twice differentiable – which is the case in this simulation – their algorithm has weaker performance than ours in this setting. Comparing our antitonic method with the VAPE algorithm by Tullii et al. (2024) (which have the same assumption on $F_0$, i.e. Lipschitzianity of $F_0$) the empirical performance of VAPE is worse than our method up to very large time horizons ($T = 8000$), achieving smaller regret.

## 6 Real Application

This study applies our method to a real data set obtained by Welltower Inc to simulate the dynamic pricing process. The dataset consists of various characteristics and the transaction price for units in the United States (see Table 2 for more details). In our experiments, we present each rental unit to the dynamic pricing algorithm in a sequential fashion to simulate the dynamic pricing game. The unique aspect of the dataset is it includes the exact transaction price, which allows us to evaluate the regret of the algorithm directly.

Table 2: Dataset description

| Variable | Description |
|---|---|
| $v_t$: **act_rate_d** | Final transaction price. |
| $x_{t,1}$: **mkt_rate_d** | Typical rate of similar unit in the primary market area. |
| $x_{t,2}$: **sqft** | Square footage of unit. |
| $x_{t,3}$: **unit_type** | Type of unit (bedroom, studio, or other). |
| $x_{t,4}$: **med_home** | Median home value of primary market area. |

This dataset doesn't contain the variable $y_t$, i.e. whether the sales occurred. Our knowledge is limited to the final transaction prices **act_rate_d**. To overcome this we make the following adjustment: at each time point $t$, we consider the transaction price **act_rate_d** as the customer valuation $v_t$, which we treat as unobserved;

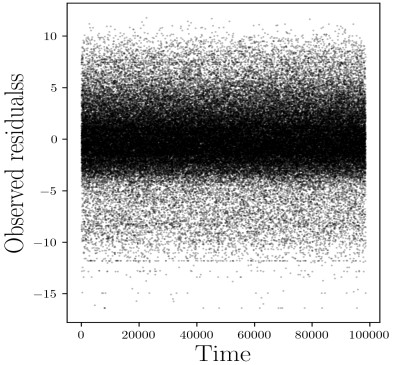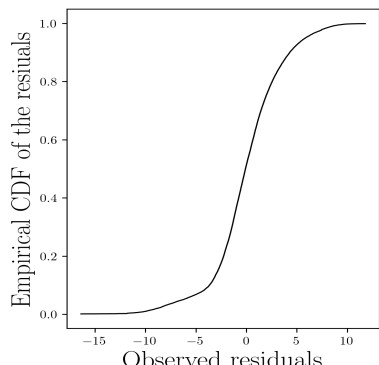

Figure 5: Residuals

Table 3: Summary Statistics

|  | mkt_rate_d | sqft | 12min_med_home_val | 20min_med_home_val | act_rate_d |
|---|---|---|---|---|---|
| min | 0.00 | 0.00 | 131131.52 | 139845.11 | 0.00 |
| 25% | 1764.12 | 334.00 | 356391.42 | 350603.75 | 139.00 |
| 50% | 2598.32 | 428.00 | 478335.94 | 481045.88 | 181.00 |
| mean | 3210.69 | 465.64 | 561100.96 | 531395.52 | 200.19 |
| 75% | 3811.40 | 546.00 | 689176.88 | 681712.58 | 233.00 |
| max | 56475.71 | 1782.00 | 1650871.95 | 1529372.81 | 1494.63 |

Table 4: Ordinary least squares: $v_t = \theta^\top x_t + z_t$

| Model: | OLS | Adj. R-squared: | 0.597 | | | |
|---|---|---|---|---|---|---|
| Df Model: | 6 | F-statistic: | 2.432e+04 | | | |
| Df Residuals: | 98552 | Prob (F-statistic): | 0.00 | | | |
| R-squared: | 0.597 | Scale: | 11.956 | | | |
| | Coef. | Std.Err. | t | P>\|t\| | [0.025 | 0.975] |
| const | 15.0307 | 0.0116 | 1299.4124 | 0.0000 | 15.0081 | 15.0534 |
| mkt_rate_d | 7.7899 | 0.0272 | 286.1036 | 0.0000 | 7.7365 | 7.8432 |
| sqft | -0.5291 | 0.0164 | -32.3591 | 0.0000 | -0.5612 | -0.4971 |
| 12min_med_home_val | 0.5431 | 0.0136 | 39.8486 | 0.0000 | 0.5164 | 0.5698 |
| unit_type_2 bed | 0.1804 | 0.0167 | 10.7981 | 0.0000 | 0.1477 | 0.2132 |
| unit_type_other | 0.1420 | 0.0175 | 8.1030 | 0.0000 | 0.1076 | 0.1763 |
| unit_type_studio | 0.0753 | 0.0218 | 3.4532 | 0.0006 | 0.0326 | 0.1181 |

a price $p_t$ is posted by the firm, which finally collects the data point $(x_t, y_t)$ where $y_t \triangleq \mathbf{1}\{v_t \geq p_t\}$. Here the vector $x_t$ contains all the variables in Table 2 except **act_rate_d**, and 1 in the first entry to account for the intercept. As we assume $v_t = \theta_0^\top x_t + z_t$ for some unknown $\theta_0$ and unknown c.d.f. $F_0$ of $z_t$, we validate this linear validation model. To this end, we perform a linear model using data $(x_t, v_t)$: the p-value of the $F$ statistics is close to 0 and the slopes of all the variables are statistically significant at a significance level of 0.05 (see Table 4). The residuals and the associated empirical distribution function (an estimate of $F_0$) are depicted in Figure 5, from which we notice that $\mathcal{U}$ is approximatively $(-17, 12)$. Moreover $p_{\min}$ and $p_{\max}$ are the maximum and minimum values of $v_t$.

Prior to implementing the methods, we conducted cross-validation to tune the UCB algorithm's parameters $\lambda$ and $C_2$, as defined in Luo et al. (2022). We searched over a grid with $(\lambda, C_2) \in \{0.1, 0.5, 1, 1.5, 2, 5\} \times \{5, 10, 15, 20, 30\}$. After selecting the optimal parameters, we ran the algorithm for each method. The initial episode length was set to $\tau_1 = 150$, with subsequent episodes doubling in length according to $\tau_k = \tau_1 2^{k-1}$, for a total of $K = 4$ episodes. Each algorithm then chooses its exploration phase according to its rule. We conducted 36 iterations, randomly shuffling the data before each run. For our algorithm, we set $\alpha = 1$.

Figure 6 showcases the (empirical) revenue $\text{Rev}(t) = \sum_{j=1}^{t} p_t y_t$, obtained using our antitonic method (blue line), the UCB method by Luo et al. (2022) (green line), the kernel method by Fan et al. (2021) (red line) and the VAPE algorithm by Tullii et al. (2024) (black line). Higher lines indicate better performance. We present three plots corresponding to different values of the smoothness parameter $m$, which affects only the kernel-based method by Fan et al. (2021). We let the antitonic, UCB, and VAPE methods remain the same across all three plots, while the kernel method's performance changes with varying $m$. Overall, our antitonic method generally outperforms the other approaches. The kernel-based method by Fan et al. (2021) also performs well and tends to improve as the smoothness parameter $m$ increases. Despite tuning its parameters, the UCB method by Luo et al. (2022) performs poorly, while as far as the VAPE algorithm by Tullii et al. (2024) shows worse performance than our method and kernel-based method by Fan et al. (2021)

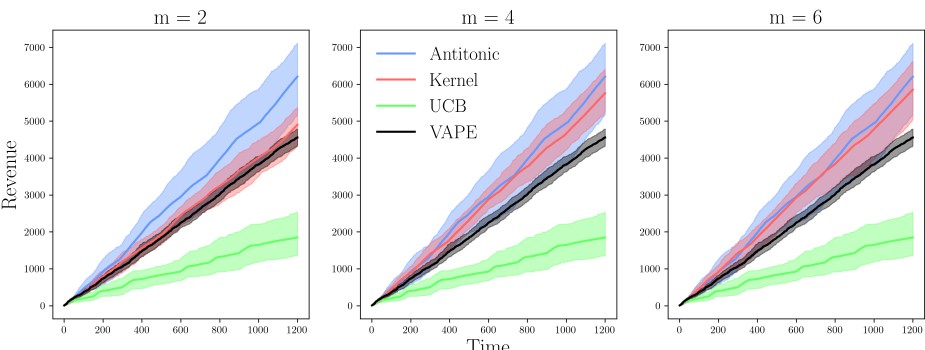

Figure 6: Revenue comparison

# 7 Conclusions

We introduced a novel method for estimating the market noise distribution $F_0$ by leveraging its natural shape constraint: monotonicity. Our analysis led to an expected upper bound on the total regret of order $\widetilde{\mathcal{O}}(T^{\nu(\alpha)}d^{\alpha/2+\alpha})$, where $\nu(\alpha)$ is defined in Equation (2), matching certain previous rates in $T$ when $\alpha = 1$ and enjoying the additional advantage of being tuning parameter-free. Compared to existing methods such as Tullii et al. (2024); Fan et al. (2021); Luo et al. (2022), our proposed algorithm shows stronger empirical performance in both simulations and real data applications.

An interesting direction for future research is the study of lower bounds on the expected regret under the Hölder condition on $S_0$ and an investigation into whether our rate matches this bound. In the special case when $\alpha = 1$ (i.e. Lipschitzianity of $F_0$), the regret lower bound of $\Omega(T^{2/3})$ established in Xu & Wang (2022), has been attained in Tullii et al. (2024). Another promising extension, particularly for practical applications, is the incorporation of optimal design strategies, as discussed in Remark 4.4. This could significantly improve the multiplicative constants in the regret, leading to more efficient algorithms.

Another promising avenue for further research is to integrate the Tullii et al. (2024) algorithm with our antitonic regression approach. The Tullii et al. (2024) method offers a better theoretical upper bound, whereas our algorithm shows stronger empirical performance. A potential strategy would be to retain the Valuation Approximation step from their Algorithm 2 while replacing the Price Elimination step with our antitonic regression procedure (as highlighted by the blue box in Algorithm 1). The final optimal price would then be computed as in Equation (5). We acknowledge that this idea is preliminary, and further investigation is required to rigorously formalize the integration and to perform a comprehensive regret analysis.

# 8 Code Availability

The codes are available at https://github.com/dbracale/DP_via_Antitonic_TMLR_2025.

# Acknowledgements

We used generative AI tools when preparing the manuscript; we remain responsible for all opinions, findings, and conclusions or recommendations expressed in the paper. This paper is based on research supported by the National Science Foundation under grants 2027737, 2113373, and 2414918.

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

# A    Missing Proofs

## A.1    Proof of Proposition 4.6

*Proof.* By Equation (7) we have $S_\theta(u) = \int S_0(u + (\theta - \theta_0)^\top x) dP_x(x)$, from which we note that $S_\theta$ is non-increasing, because $S_0$ is non-increasing. Moreover if $S_0$ is $\alpha$-Hölder,

$$
\begin{aligned}
|S_\theta(u) - S_\theta(v)| &= \int |S_0(u + (\theta - \theta_0)^\top x) - S_0(v + (\theta - \theta_0)^\top x)| dP_x(x) \\
&\leq \int C_1 |u - v|^\alpha dP_x(x) = C_1 |u - v|^\alpha,
\end{aligned}
$$

and

$$
\begin{aligned}
|S_\theta(u) - S_0(u)| &= \int |S_0(u + (\theta - \theta_0)^\top x) - S_0(u)| dP_x(x) \\
&\leq \int C_1 |(\theta - \theta_0)^\top x|^\alpha dP_x(x) \\
&\leq C_1 R_{\mathcal{X}}^\alpha \|\theta - \theta_0\|_2^\alpha,
\end{aligned}
$$

where in the last inequality we used Cauchy-Scwartz and that $\|x\|_2 \leq R_{\mathcal{X}}$. $\qquad\square$

## A.2    Proof of Theorem 4.8

We first need to Lemmas: Lemma A.1 and Lemma A.2.

**Lemma A.1.** *Let*

$$
\bar{S}_{rs}(\theta) \triangleq \frac{1}{o_{rs}} \sum_{j=r}^s o_j S_\theta(w_j),
$$

*and*

$$
M_n(\theta) \triangleq \max_{1 \leq r \leq s \leq m} o_{rs}^{1/2} |\widehat{y}_{rs} - \bar{S}_{rs}(\theta)|.
$$

*Then for any constant $D > 1$,*

$$
\mathbb{P}\left( M_n(\theta) \leq (D \log n)^{1/2} \right) \leq 1 - \left( \tfrac{n+1}{n^D} \right)^2.
$$

*Proof.* First, the by Hoeffding's inequality, since $y_j$ are independent random variables taking values $\{0, 1\}$ with mean $S_\theta(w_t)$, for every $\eta > 0$ we have

$$
\mathbb{P}\left[ \sqrt{o_{rs}} |\widehat{y}_{sr} - \bar{S}_{sr}(\theta)| \geq \eta \right] \leq 2e^{-2\eta^2}.
$$

Note that $M_n$ is the maximum of the $\binom{m+1}{2}$ quantities

$$
o_{rs}^{1/2} |\widehat{y}_{rs} - \bar{S}_{rs}(\theta)|.
$$

Consequently,

$$
\begin{aligned}
\mathbb{P}\left( M_n(\theta) \geq \eta_n \right) &\leq \sum_{1 \leq r \leq s \leq m} \mathbb{P}\left( o_{rs}^{1/2} |\widehat{y}_{rs} - \bar{S}_{rs}(\theta)| \geq \eta_n \right) \\
&\leq 2 \binom{m}{2} \exp\left( -2\eta_n^2 \right) \\
&\leq \exp\left( 2\log(n+1) - 2\eta_n^2 \right) \\
&\leq \exp\left( 2\log((n+1)/n^D) \right) = \left( \tfrac{n+1}{n^D} \right)^2,
\end{aligned}
$$

for arbitrary $\eta_n \geq 0$. But the right hand side converges to zero as $n \to \infty$ if $\eta_n = (D \log n)^{1/2}$ for some $D > 1$. $\qquad\square$

Before proceeding with the technical Lemma A.2, let's define

$$\rho_n \triangleq \frac{\log n}{n},$$

and $\lambda(\cdot)$ the Lebesgue measure, and denote by $P_n(\cdot)$ the empirical measure of the design points $w_t$, that means

$$P_n(B) \triangleq \frac{1}{n} \# \{t \in \mathcal{T} : w_t \in B\} \quad \text{for } B \subset \mathcal{U}.$$

**Lemma A.2.** *Let $w_1, w_2, \ldots, w_n$ i.i.d. points with density $f_w$ that satisfies $\inf_{u \in \mathcal{U}} f_w(u) \geq C_2$ for some universal constant $C_2 > 0$ (which is the case for the uniform distribution), then for a given constant $\kappa > 0$, and for any $\gamma > 2$, there exists $n_0 = n_0(\gamma, \kappa, \alpha) \in \mathbb{N}$ and a sequence $\epsilon_n = \epsilon_n(\gamma, \kappa, \alpha) > 0$, $\epsilon_n \to 0$ such that*

$$\mathbb{P}(A_{n,\gamma}) > 1 - \frac{1}{2(n+2)^{\gamma-2}}, \quad n \geq n_0$$

*where $A_{n,\gamma}$ is the event*

$$\inf \left\{ \frac{P_n(\mathcal{U}_n)}{\lambda(\mathcal{U}_n)} : \mathcal{U}_n \subset \mathcal{U}, \lambda(\mathcal{U}_n) \geq \delta_n \triangleq \kappa \rho_n^{1/(2\alpha+1)} \right\} \geq C_2(1 - \epsilon_n).$$

*Proof.* This is immediately derived from the proof of the more general result by Mösching & Dümbgen (2020, Section 4.3) which can be stated as follows: let $\delta_n > 0$ such that $\delta_n \to 0$ while $n\delta_n / \log(n) \to \infty$ (as $n \to \infty$). Then for every $\gamma > 2$, there exists $n_0 = n_0(\gamma, \delta_n)$ and $\epsilon_n = \epsilon_n(\gamma, \delta_n) > 0$, $\epsilon_n \to 0$ such that

$$\mathbb{P}\left( \inf \left\{ \frac{P_n(\mathcal{U}_n)}{P(\mathcal{U}_n)} : \mathcal{U}_n \subset \mathcal{U}, P(\mathcal{U}_n) \geq \delta_n \right\} \geq 1 - \epsilon_n \right) > 1 - \frac{1}{2(n+2)^{\gamma-2}}, \quad n \geq n_0,$$

where $P(\cdot)$ is the probability measure of the design points $w_t$, that is

$$P(B) \triangleq \int_B f_w(w) dw, \quad \text{for } B \subset \mathcal{U},$$

and

$$\epsilon_n \triangleq \max \left( c_n/\delta_n, \sqrt{2c_n/\delta_n} \right) + (n\delta_n)^{-1} \to 0,$$

where $c_n \triangleq \gamma \log(n+2)/(n+1)$. The value $n_0$ is the smallest integer $n$ that satisfies $\epsilon_n < 1$. $\square$

Now we prove Theorem 4.8.

*Proof.* Let $n$ be sufficiently large so that $\mathcal{U}_n \neq \emptyset$ and such that the event $A_{n,\gamma}$ in Lemma A.2 occurs. Since $f_w$ is the uniform distribution, the value $C_2$ defined in Lemma A.2 corresponds to $1/|\mathcal{U}|$. For $u \in \mathcal{U}_n$ the indices

$$r(u) \triangleq \min \{j \in \{1, \ldots, m\} : u_j \geq u - \delta_n\},$$
$$j(u) \triangleq \max \{j \in \{1, \ldots, m\} : u_j \leq u\},$$

are well-defined, because $[u - \delta_n, u]$ is a subinterval of $I$ of length $\delta_n$. Note that by Lemma A.2 this interval contains at least one observation $u_j$. Moreover,

$$r(u) \leq j(u),$$
$$u - \delta_n \leq u_{r(u)} \leq u_{j(u)} \leq u,$$
$$o_{r(u)j(u)} = o_n([u - \delta_n, u]) \geq C_2(1 - \epsilon_n) n\delta_n,$$

where $\epsilon_n$ is defined as in Lemma A.2. Consequently, with $M_n(\theta)$ as in Lemma A.1, we have

$$
\begin{aligned}
\widehat{S}_\theta(u) - S_\theta(u) &\leq \widehat{S}_\theta(u_{j(u)}) - S_\theta(u) \\
&= \min_{r \leq j(u)} \max_{s \geq j(u)} \widehat{y}_{rs} - S_\theta(u) \\
&\leq \max_{s \geq j(u)} \widehat{y}_{r(u)s} - S_\theta(u) \\
&\leq o_{r(u)j(u)}^{-1/2} M_n(\theta) + \max_{s \geq j(u)} \bar{S}_{r(u)s} - S_\theta(u) \\
&\leq (C_2(1-\epsilon_n)n\delta_n)^{-1/2} M_n(\theta) + S_\theta(u_{r(u)}) - S_\theta(u) \\
&\leq (C_2(1-\epsilon_n)n\delta_n)^{-1/2} M_n(\theta) + C_1\delta_n^\alpha.
\end{aligned}
$$

In the first step, we used antitonicity of $u \mapsto \widehat{S}_\theta(u)$, and in the second last step we used antitonicity of $u \mapsto S_\theta(u)$, and the last step utilizes that by Assumption 4.7. But on the event $\{M_n(\theta) \leq (D\log n)^{1/2}\}$, the previous considerations implies that

$$
\sup_{u \in \mathcal{U}_n} (\widehat{S}_\theta(u) - S_\theta(u)) \leq (C_2(1-\epsilon_n)n\delta_n)^{-1/2} (D\log n)^{1/2} + C_1\delta_n^\alpha = C\rho_n^{\alpha/(2\alpha+1)},
$$

where $C = \sqrt{\kappa D/C_2} + C_1\kappa^\alpha$, and we recall that $C_2 = 1/|\mathcal{U}|$ and $D$ is any real value strictly greater than 1. But $\sup_{u \in \mathcal{U}_n} (S_\theta(u) - \widehat{S}_\theta(u)) \leq C\rho_n^{\alpha/(2\alpha+1)}$ happens in $A_{n,\gamma} \cap \{M_n(\theta) \leq (D\log n)^{1/2}\}$ which has probability

$$
\begin{aligned}
\mathbb{P}(A_{n,\gamma} \cap \{M_n(\theta) \leq (D\log n)^{1/2}\}) &= 1 - \mathbb{P}(A_{n,\gamma}^c \cup \{M_n(\theta) \geq (D\log n)^{1/2}\}) \\
&\geq 1 - \mathbb{P}(A_{n,\gamma}^c) - \mathbb{P}(M_n(\theta) \geq (D\log n)^{1/2}) \\
&= \mathbb{P}(A_{n,\gamma}) + \mathbb{P}(M_n(\theta) \leq (D\log n)^{1/2}) - 1 \\
&\geq 1 - \frac{1}{2(n+2)^{\gamma-2}} - \left(\frac{n+1}{n^D}\right)^2 \\
&\geq 1 - \frac{1}{(n+2)^{\gamma-2}} \geq 1 - \frac{1}{n^{\gamma-2}},
\end{aligned}
$$

where we used that by Lemma A.1, for any fixed $D > 1$ we have $\mathbb{P}\left(M_n(\theta) \leq (D\log n)^{1/2}\right) \geq 1 - (\frac{n+1}{n^D})^2$ and by Lemma A.2 for any $\gamma > 2$ we have $\mathbb{P}\left(A_{n,\gamma}\right) > 1 - \frac{1}{2(n+2)^{\gamma-2}}$. The last two inequalities come from choosing $\gamma = D$ sufficiently large.

Analogously one can show that on $\{M_n \leq (D\log n)^{1/2}\}$,

$$
\sup_{u \in \mathcal{U}_n} (S_\theta(u) - \widehat{S}_\theta(u)) \leq (n\delta_n)^{-1/2} (D\log n)^{1/2} + C_1\delta_n^\alpha = C\rho_n^{\alpha/(2\alpha+1)},
$$

with the same constant $C$ and with the same probability tail. $\qquad\square$

### A.3 Proof of Theorem 4.10

Fix $k \geq 2$ and define $n_k = |I_k|$ and $\widetilde{n}_k = |\widetilde{I}_k|$ and $a_k = |E_k| = n_k + \widetilde{n}_k$. Let $S_0(p \mid x) \triangleq S_0(p - \theta_0^\top x)$ and $\widehat{S}_k(p \mid x) \triangleq \widehat{S}_k(p - \widehat{\theta}_k^\top x)$. For the exploration phase $\mathbb{E}[\sum_{t \in E_k} r_t(p_t^*) - r_t(p_t)] \leq p_{\max}|E_k| \lesssim |E_k|$. Now fix $t \in E_k'$

$$
\begin{aligned}
&r_t(p_t^*) - r_t(p_t) \\
&= p_t^* S_0(p_t^* \mid x_t) - p_t S_0(p_t \mid x_t) \\
&= \left\{p_t^* S_0(p_t^* \mid x_t) - p_t^* \widehat{S}_k(p_t^* \mid x_t)\right\} + \underbrace{\left\{p_t^* \widehat{S}_k(p_t^* \mid x_t) - p_t \widehat{S}_k(p_t \mid x_t)\right\}}_{\leq 0 \text{ by Equation (6)}} + \left\{p_t \widehat{S}_k(p_t \mid x_t) - p_t S_0(p_t \mid x_t)\right\} \\
&\leq p_{\max}\left|S_0(p_t^* \mid x_t) - \widehat{S}_k(p_t^* \mid x_t)\right| + p_{\max}\left|\widehat{S}_k(p_t \mid x_t) - S_0(p_t \mid x_t)\right| \\
&= R_{k,t}(p_t^*) + R_{k,t}(p_t),
\end{aligned}
$$

where $R_{k,t}(q) \triangleq |\widehat{S}_k(q - \widehat{\theta_k}^\top x_t) - S_0\left(q - \theta_0^\top x_t\right)|$ for $q \in \{p_t^*, p_t\}$, $t \in E_k'$.

By Lemma 4.11 there exists $k_0$ such that for $k \geq k_0$, $\mathbb{E}(R_{k,t}(q)) \lesssim \left(\log \widetilde{n}_k / \widetilde{n}_k\right)^{\alpha/2\alpha+1} + \left(d \log n_k / n_k\right)^{\alpha/2}$ for $q \in \{p_t^*, p_t\}$ with $t \in E_k'$. Summing up for all $t \in E_k'$, yields that

$$\mathbb{E}\left[\sum_{t \in E_k'} r_t(p_t^*) - r_t(p_t)\right] \lesssim |E_k'| \left[\left(\log \widetilde{n}_k / \widetilde{n}_k\right)^{\alpha/2\alpha+1} + \left(d \log n_k / n_k\right)^{\alpha/2}\right].$$

Merging with the exploration phase of episode $k$ we get

$$\mathbb{E}\left[\sum_{t \in \mathcal{J}_k} r_t(p_t^*) - r_t(p_t)\right] \lesssim |E_k| + |E_k'| \left[\left(\log \widetilde{n}_k / \widetilde{n}_k\right)^{\alpha/2\alpha+1} + \left(d \log n_k / n_k\right)^{\alpha/2}\right].$$

Using that $n_k = \widetilde{n}_k = \frac{1}{2} a_k = \frac{1}{2} |E_k| = \frac{1}{2} d^\xi (\tau_1 2^{k-1})^\nu \propto d^\xi 2^{k\nu}$ for $\xi, \nu > 0$ to be determined such that they minimize the total regret, and that $|E_k'| \leq |\mathcal{J}_k| = \tau_1 2^{k-1} \propto 2^k$ we get that the RHS of the last inequality is

$$\mathbb{E}\left[\sum_{t \in \mathcal{J}_k} r_t(p_t^*) - r_t(p_t)\right] \lesssim d^\xi 2^{k\nu} + 2^k \left[\left(\frac{\log(d^\xi 2^{k\nu})}{d^\xi 2^{k\nu}}\right)^{\alpha/2\alpha+1} + \left(\frac{d \log(d^\xi 2^{k\nu})}{d^\xi 2^{k\nu}}\right)^{\alpha/2}\right]$$

$$\lesssim d^\xi 2^{k\nu} + 2^k \left(\frac{\log(d^\xi 2^{k\nu})}{d^\xi 2^{k\nu}}\right)^{\alpha/2\alpha+1} + 2^k \left(\frac{d \log(d^\xi 2^{k\nu})}{d^\xi 2^{k\nu}}\right)^{\alpha/2}$$

$$\lesssim d^\xi 2^{k\nu} + d^{-\frac{\xi\alpha}{2\alpha+1}} 2^{k(1-\frac{\nu\alpha}{2\alpha+1})} [k + \log(d)]^{\frac{\alpha}{2\alpha+1}} + d^{\frac{\alpha}{2}(1-\xi)} 2^{k(1-\frac{\nu\alpha}{2})} [k + \log(d)]^{\frac{\alpha}{2}}.$$

The exponents of the factor $d$ are $\xi, -\frac{\xi\alpha}{2\alpha+1}$ and $\frac{\alpha}{2}(1-\xi)$. As the second exponent is always negative we equalize the first and the third exponent, i.e. $\xi = \frac{\alpha}{2}(1 - \xi)$ to get $\xi^* = \frac{\alpha}{\alpha+2}$. The exponents of the exponential factor $2^k$ are $\nu$, $1 - \frac{\nu\alpha}{2\alpha+1}$ and $1 - \frac{\nu\alpha}{2}$. Equalizing the first two factors, we get $\nu^* = \frac{2\alpha+1}{3\alpha+1}$, however $\nu^* > (1 - \frac{\nu^*\alpha}{2})$ for $\alpha > 1/2$, is equal for $\alpha = 1/2$ and less for $\alpha < 1/2$. Then for $\alpha \geq 1/2$ we equalizing the first and last factors, obtaining $\nu = 1 - \frac{\nu\alpha}{2}$ to get $\nu^* = \frac{2}{2+\alpha}$.

**Case $\alpha > 1/2$.** The expected regret in episode $k$, $\mathbb{E}\left[\sum_{t \in \mathcal{J}_k} r_t(p_t^*) - r_t(p_t)\right]$ is upper bounded by

$$2^{k\frac{2\alpha+1}{3\alpha+1}}(d^{\frac{\alpha}{\alpha+2}} + d^{-\frac{\alpha^2}{(2\alpha+1)(\alpha+2)}} [k + \log(d)]^{\frac{\alpha}{2\alpha+1}} + d^{\frac{\alpha}{\alpha+2}} [k + \log(d)]^{\frac{\alpha}{2}}) \lesssim 2^{k\frac{2\alpha+1}{3\alpha+1}} d^{\frac{\alpha}{\alpha+2}} [k + \log(d)]^{\frac{\alpha}{2}},$$

where we used that $\frac{\alpha}{2\alpha+1} < \frac{\alpha}{2}$ for $\alpha \in (1/2, 1]$. Putting together the phases we get

$$R(T) = \mathbb{E}\left[\sum_{k=k_0}^{K} \sum_{t \in \mathcal{J}_k} r_t(p_t^*) - r_t(p_t)\right] \lesssim 2^{K\frac{2\alpha+1}{3\alpha+1}} d^{\frac{\alpha}{\alpha+2}} [K + \log(d)]^{\frac{\alpha}{2}} \lesssim T^{\frac{2\alpha+1}{3\alpha+1}} d^{\frac{\alpha}{\alpha+2}} \log^{\frac{\alpha}{2}}(dT),$$

where we used that $K = \lceil \log(T/\tau_1) + 1 \rceil$.

**Case $\alpha \leq 1/2$.** The expected retreat in episode $k$, $\mathbb{E}\left[\sum_{t \in \mathcal{J}_k} r_t(p_t^*) - r_t(p_t)\right]$, is upper bounded by

$$2^{k\frac{2}{2+\alpha}}(d^{\frac{\alpha}{\alpha+2}} + d^{-\frac{\alpha^2}{(2\alpha+1)(\alpha+2)}} [k + \log(d)]^{\frac{\alpha}{2\alpha+1}} + d^{\frac{\alpha}{\alpha+2}} [k + \log(d)]^{\frac{\alpha}{2}}) \lesssim 2^{k\frac{2}{2+\alpha}} d^{\frac{\alpha}{\alpha+2}} [k + \log(d)]^{\frac{\alpha}{2\alpha+1}},$$

where we used that $\frac{\alpha}{2\alpha+1} \geq \frac{\alpha}{2}$ for $\alpha \in (0, 1]$. Putting together the phases we get

$$R(T) = \mathbb{E}\left[\sum_{k=k_0}^{K} \sum_{t \in \mathcal{J}_k} r_t(p_t^*) - r_t(p_t)\right] \lesssim 2^{K\frac{2}{2+\alpha}} d^{\frac{\alpha}{\alpha+2}} [K + \log(d)]^{\frac{\alpha}{2\alpha+1}} \lesssim T^{\frac{2}{2+\alpha}} d^{\frac{\alpha}{\alpha+2}} \log^{\frac{\alpha}{2\alpha+1}}(dT),$$

where we used that $K = \lceil \log(T/\tau_1) + 1 \rceil$, which concludes the proof.

### A.4 Proof of Lemma 4.11

Let $n_k = |I_k|$ and $\widetilde{n}_k = |\widetilde{I}_k|$ and $t \in E'_k$. Define the event $\mathcal{E}_k = \{\|\widehat{\theta}_k - \theta_0\| \leq R_{n_k}\}$ where we recall

$$R_{n_k} \propto \sqrt{\frac{d \log n_k}{n_k}},$$

as defined in Lemma 4.3, and

$$R_{k,t}(q) = |\widehat{S}_k(q - \widehat{\theta}_k^\top x_t) - S_0(q - \theta_0^\top x_t)|, \quad q \in \{p_t, p_t^*\}.$$

Recall that

$$\widehat{\theta}_k = \text{OLS}\{(x_t, y_t)\}_{t \in I_k}, \quad \widehat{S}_k = \text{Antitonic}\{(w_t, y_t)\}_{t \in \widetilde{I}_k},$$

and define $S_k = S_{\widehat{\theta}_k}$, where by definition in Equation (7)

$$S_k(u) = S_{\widehat{\theta}_k}(u) = \mathbb{E}_x[S_0(u + (\widehat{\theta}_k - \theta_0)^\top x)].$$

Now let $q \in \{p_t, p_t^*\}$ for some $t \in E'_k$. We can write

$$R_{k,t}(q) = R_{k,t}(q)\mathbb{I}(\mathcal{E}_k) + R_{k,t}(q)\mathbb{I}(\mathcal{E}_k^c).$$

**Analyzing the $R_{k,t}(q)\mathbb{I}(\mathcal{E}_k^c)$:**

By Lemma 4.3 we have $\mathbb{E}[R_{k,t}(q)\mathbb{I}(\mathcal{E}_k^c)] \leq 2\mathbb{P}(\mathcal{E}_k^c) = Q_{n_k} = 2e^{-c_1 c_{\min}^2 n_k/16} + \frac{2}{n_k}$.

**Analyzing the $R_{k,t}(q)\mathbb{I}(\mathcal{E}_k)$:**

$R_{k,t}(q)$ is less or equal than two times

$$\underbrace{|\widehat{S}_k(q - \widehat{\theta}_k^\top x_t) - S_k(q - \widehat{\theta}_k^\top x_t)|\mathbb{I}(\mathcal{E}_k)}_{=A} + \underbrace{|S_k(q - \widehat{\theta}_k^\top x_t) - S_k(q - \theta_0^\top x_t)|\mathbb{I}(\mathcal{E}_k)}_{=B} + \underbrace{|S_k(q - \theta_0^\top x_t) - S_0(q - \theta_0^\top x_t)|\mathbb{I}(\mathcal{E}_k)}_{=C} \tag{9}$$

**Analyzing $A$ on $\mathcal{E}_k$:** Define the event $\mathcal{S}_k = \{\sup_{u \in \mathcal{U}} |\widehat{S}_k(u) - S_k(u)| \leq C\rho_{n_k}^{\alpha/(2\alpha+1)}\}$, where $\rho_n = \log(n)/n$. For $n_k$ sufficiently large, by Theorem 4.8 we have that

$$\mathbb{E}(A) = \mathbb{E}(A\mathbb{I}(\mathcal{S}_k, \cap \mathcal{E}_k)) + \mathbb{E}(A\mathbb{I}(\mathcal{S}_k^c \cap \mathcal{E}_k))$$

$$\leq \mathbb{E}\left(\sup_{u \in \mathcal{U}} |\widehat{S}_k(u) - S_k(u)|\mathbb{I}(\mathcal{S}_k \cap \mathcal{E}_k)\right) + 2\mathbb{P}(\mathcal{S}_k^c)\mathbb{P}(\mathcal{E}_k)$$

$$\lesssim C\left(\frac{\log \widetilde{n}_k}{\widetilde{n}_k}\right)^{\alpha/(2\alpha+1)}\mathbb{P}(\mathcal{S}_k \cap \mathcal{E}_k) + 2\mathbb{P}(\mathcal{S}_k^c)$$

$$\lesssim \left(\frac{\log \widetilde{n}_k}{\widetilde{n}_k}\right)^{\alpha/(2\alpha+1)} + 2\frac{1}{\widetilde{n}_k^{\gamma-2}}$$

$$\lesssim \left(\frac{\log \widetilde{n}_k}{\widetilde{n}_k}\right)^{\alpha/(2\alpha+1)},$$

where we chose $\gamma \geq 3$.

**Analyzing $B$ on $\mathcal{E}_k$:** By Proposition 4.6, $S_k$ is $\alpha$-Hölder, then $\mathbb{E}[B\mathbb{I}(\mathcal{E}_k)] \lesssim \|\widehat{\theta}_k - \theta_0\|_2^\alpha \leq R_{n_k}^\alpha$.

**Analyzing $C$ on $\mathcal{E}_k$:** By Proposition 4.6 we have $|S_k(u) - S_0(u)|\mathbb{I}(\mathcal{E}_k) \lesssim \|\widehat{\theta}_k - \theta_0\|_2^\alpha \leq R_{n_k}^\alpha$.

**Combining the terms $R_{k,t}(q)\mathbb{I}(\mathcal{E}_k^c)$ and $R_{k,t}(q)\mathbb{I}(\mathcal{E}_k)$ from Equation (9):** we get

$$\sup_q R_{k,t}(q) \lesssim \left(\frac{\log \widetilde{n}_k}{\widetilde{n}_k}\right)^{\alpha/(2\alpha+1)} + \left(\frac{d \log n_k}{n_k}\right)^{\alpha/2}.$$

## B  Additional Plots of Section 5.1

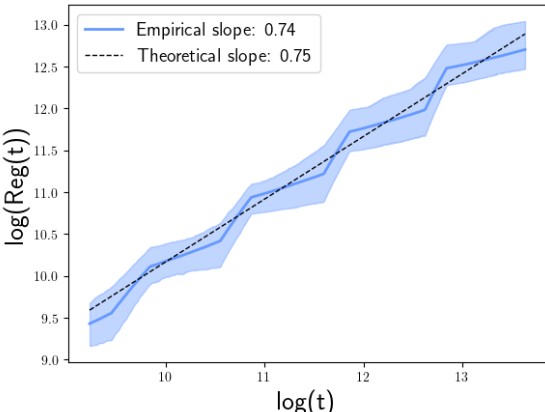

Figure 7: This plot was generated using as true $F_0$ the one considered in Fan et al. (2021) with density $f_0(z) = 6\left(\frac{1}{4} - z^2\right)\mathbf{1}\{z \in (-1/2, 1/2)\}$. We repeated the simulation 36 times and the corresponding 95% confidence interval. The plot is in $\log_2$-$\log_2$ scale to show the regret rate (empirical slope): a slope of $\eta$ indicated an $\mathcal{O}(T^\eta)$ regret. The black dashed line corresponds to our theoretical regret upper bound of 3/4. The estimated slope is very close to that value.

