# OpenReview forum: "Dynamic Pricing in the Linear Valuation Model using Shape Constraints"
_TMLR — Accepted by TMLR_

### Review · Reviewer_enYp · 2025-02-25

**Summary Of Contributions:**

The paper studies the problem of designing regret minimizers for dynamic pricing. A fundamental feature is that the regret minimizer can exploit contextual data.  In particular, at each round the market value depends linearly on an observed context and some noise. The authors achieve sublinear regret bounds assuming that the noise is Holder continuous. This is an improvement over previous works, that assume (at least) Lipschitz noise. The theoretical analysis is complemented by an empirical analysis that shows that the algorithm performs well in practice.

**Audience:**

Yes

**Broader Impact Concerns:**

None.

**Claims And Evidence:**

Yes

**Requested Changes:**

Clarify the relation with previous works.

**Strengths And Weaknesses:**

Strengths: the paper provides the first theoretical results for Holder noise. The proposed algorithm has good empirical performance.

Weaknesses: the contribution of the paper and its relation with previous works is unclear on multiple aspects.

1) The comparison with the work of Tullii at al. (2024) sounds misleading. While I agree that the choice of the parameter in the paper leads to a $(1-1/log(T))$ probability, it seems more like a "typo" to me. Indeed, $\alpha$ appears only in logarithmic components of the high-probability regret bound and can be reduced to any term $T^c$, with c constant, at the cost of a negligible logarithmic increase in the high-probability regret". To conclude, their proof is essentially correct and I think the authors should consider their bound sound. Am I missing something?

2) The paper is based on some assumptions that are neither common to all previous works, nor without loss of generality. Moreover, some assumptions are not clearly stated in the statements. For instance, it is not clear that you are assuming that $F_0$ is not decreasing.
To conclude, in the Lipschitz setting you are achieving worst (or at best, no better) regret bound with stronger assumptions. You should better highlight, e.g., in the introduction, that in this setting the regret of your algorithm is $O(T^{3/4})$.

To conclude, the only theoretical  improvement over previous woks seems the ability to handle Holder noise.

---

> ### Author Response · Authors · 2025-03-01
>
> We thank the reviewer for their valuable feedback and address the points raised below:
>
> **Comparison with the work of Tullii et al. (2024).** Between the time we submitted the paper and when we received the review, we got in touch with the authors of Tulli et al. (2024) and they pointed out the same to us. We have therefore changed our discussion of Tulli et. al. in the revision: at the end of Section 1.1, Table 1, Section 5.2, and Section 7.
>
> **Additional Revision Updates.** We have explicitly clarified before equation (5), and after equation (7) that $S_0 = 1 - F_0$, where $F_0$ represents the noise distribution, and thus, $S_0$ is a survival function and is therefore necessarily non-increasing. Additionally, we have emphasized after equation (2) that our algorithm's regret scales as $T^{3/4}$ when $\alpha=1$.

---

> > ### Comment · Action_Editor_nsUd · 2025-03-07
> > **On comparison with Tullii at al. (2024)**
> >
> > Dear Reviewer enYp,
> >
> > If I understand correctly, your claim is that the regret bound in this paper is a high-probability bound of probability $1 - (\log T)^{-1}$, which results in $O(T/(\log T))$ regret bound in expectation, right?
> >
> > Best,
> > AE

---

### Review · Reviewer_S38K · 2025-02-27

**Summary Of Contributions:**

The paper studies the dynamic pricing problem in an online market and formulates it as a regret minimization problem. The main contribution is extending the noise distribution in the feedback from Lipschitz to $\alpha$-Hölder continuity while maintaining the same linear model commonly studied in practice. Within this new setting, the paper introduces novel algorithms and provides regret upper bounds for these algorithms. Although the paper claims the novelty of introducing a new analytical technique, this is unclear based on the current writing. A clearer demonstration of this technique, if it exists, would be beneficial.

**Audience:**

Yes

**Broader Impact Concerns:**

To the best of my knowledge, there are no concerns regarding the broader impact.

**Claims And Evidence:**

Yes

**Requested Changes:**

First, please refer to the weaknesses section for suggested changes.

It is also important to specify the necessity of having $p_{min}$ and $p_{max}$.

Additionally, it would be helpful if the paper included a discussion on the technical challenges introduced by $\alpha$-Hölder continuity compared to the traditional Lipschitz setting. Whether such an extension is a straightforward step based on the existing results requires further justifications.

Likewise, the technical novelty remains unclear, especially in the absence of any intuition or proof sketch for the theoretical results. It is not evident what the main technical contribution is with $\alpha$-Hölder continuity. In fact, it is not fully convincing whether considering $\alpha$-Hölder continuity has a significant impact. I believe that including proof sketches would be very beneficial. At least for now, it is unclear how the results/proof correspond to the claimed contribution (3) in Section 1, i.e. in terms of the new technique.

Furthermore, it remains unclear how to choose $\tau_1$, given that the paper claims a contribution of being tuning-parameter-free. Including a discussion or insight on this aspect would be helpful.

**Strengths And Weaknesses:**

Strengths:

The extension from Lipschitz to $\alpha$-Hölder continuity represents a novel contribution.

The paper is clearly written and easy to follow; it has been a pleasure reading the paper.

The method is examined both theoretically and practically, and also applied to real-world datasets.

The regret upper bound depends on $\alpha$, which reflects the additional complexity of the considered problem setting herein. Moreover, the regret upper bound matches existing work when $\alpha = 1$, i.e., the well-studied Lipschitz setting.

Weaknesses:

It is not completely clear what motivates the consideration of $\alpha$-Hölder continuity, especially in the dynamic online market scenario

There is inconsistency in the naming of benchmarks in the numerical experiments, e.g., Fan Method versus Kernel Method.

A regret bound in expectation does not necessarily imply a regret bound in high probability. However, in existing work, a regret bound in high probability does not necessarily imply a regret bound in expectation.

It is unclear about whether Assumption 4.10 is too strong, since assuming the existence of such minimizers can be impractical.

In a linear setting, the context $x_t$ is assumed to be i.i.d. distributed, which can be a strong assumption. In contrast, in existing work—for example, in linear contextual bandits—the context is arbitrary.

There seems to be no regret lower bound that matches the regret upper bounds. Without such a bound, one cannot claim the optimality of the method.

---

> ### Author Response · Authors · 2025-03-10
>
> We thank the reviewer for their valuable feedback and address the points raised below:
>
> - **Naming of benchmarks in the numerical experiments.** In the revision we named the benchmark consistently in all the numerical experiments.
>
> - **Assumption 4.10 Removal.** In the revision we have removed Assumption 4.10, as it is not necessary for the derivation of our results.
>
> - **iid context assumption.** In the revision we have expanded our discussion regarding the i.i.d. assumption on the context vectors immediately following Assumption 4.2: in our paper, we have worked under a set of standard assumptions used by a number of previous authors.
>
> - **Regret Bounds and Optimality.** We clarify that we do not claim the optimality of our algorithm; in the Conclusions section we explicitly state that deriving a matching lower bound for the $\alpha \in (0,1)$ case remains an important direction for future research.
>
> - **Necessity of having $[p_{\min}, p_{\max}]$.** The compactness of the interval where the prices are set is not only realistic but also useful for estimating $\theta_0$ in the exploration phase (recall that we sample uniform prices in $[p_{\min},p_{\max}]$ for this purpose).
>
> - **Motivation and Novelty of $\alpha$-Holder continuity.** The technical derivation with $\alpha$-Holder continuity is a natural extension from the Lipschitz case (which has not been previously studied when $\alpha \neq 1$) and does not present a dramatically new challenge; however, its inclusion broadens the applicability of our method to scenarios where the smoothness assumption is weaker. As far as our contribution 3) is concerned, this is a contribution to the nonparametric shape-constrained literature, namely, we provide a concentration inequality for the isotonic estimator under any Holder continuity parameter $\alpha \in (0,1]$.
>
> - **Proof Sketch.** In the revision we have added a proof sketch of the main Theorem 4.10.
>
> - **Discussion on the parameter $\tau_1$.** Although $\tau_1$ is generally considered a non-tuning parameter --- typically set to $1$ in dynamic pricing algorithms that use the doubling trick (see, e.g., [1,2,3]) --- in the revision we have included an additional simulation in Section 5.1 (see Remark 5.1) to demonstrate the robustness of our algorithm with respect to this parameter.
>
> ## References
> [1] Adel Javanmard and Hamid Nazerzadeh. "Dynamic pricing in high-dimensions". Journal of Machine Learning
> Research, 20(9):1–49, 2019.
>
> [2] Javanmard, Adel, Hamid Nazerzadeh, and Simeng Shao. "Multi-product dynamic pricing in high-dimensions with heterogeneous price sensitivity." 2020 IEEE International Symposium on Information Theory (ISIT). IEEE, 2020.
>
> [3] Javanmard, Adel, Jingwei Ji, and Renyuan Xu. "Multi-Task Dynamic Pricing in Credit Market with Contextual Information." arXiv preprint arXiv:2410.14839 (2024).

---

### Review · Reviewer_q6TL · 2025-03-03

**Summary Of Contributions:**

This paper develops a novel algorithm for dynamic pricing using a linear valuation function under three mild assumptions: (i) bounded parameters, (ii) i.i.d. contexts, and (iii) an α-Hölder continuous noise distribution.

Key contributions include:
- A novel tuning parameter-free algorithm that relies on weaker assumptions about the market noise distribution compared to existing methods.
- An analysis of the algorithm’s regret upper bound with a new concentration inequality.
- Numerical experiments with both simulated and real-world data. The proposed algorithm performs well on both of them.

**Audience:**

Yes

**Broader Impact Concerns:**

There are no broader impact concerns associated with this work.

**Claims And Evidence:**

Yes

**Requested Changes:**

Please expand the discussion regarding the i.i.d. assumption on the context vectors. For instance, is this assumption critical to relaxing the Lipschitz condition as required in Tullii et al. (2024)? What challenges might arise when dealing with non-i.i.d. contexts while employing the UCB/LCB-based algorithm from Tullii et al. (2024) or when potentially integrating it with the proposed antitonic regression?

In addition, it would be helpful if there are more discussions about the challenges of extending the current algorithm to accommodate non-linear valuation functions.

Minor Suggestions:

- Potential typo in abstract: “Previous works have addressed the challenge of unknown market noise distribution $F_0$” instead of “$F$”.
- Ensure that $S_0​$ is formally defined.

**Strengths And Weaknesses:**

Strengths:

- The paper is well written and easy to follow, with detailed remarks that clarify the algorithm design and the associated proof sketches.
- The idea of isotonic regression is new to the literature of dynamic pricing.
- The removal of tuning parameters in the existing algorithm simplifies practical implementation.
- Good numerical performance on both simulated and real-world data.

Weaknesses:
- The paper assumes a linear valuation function, whereas Tullii et al. (2024) have addressed non-linear cases.
- The assumption that contexts are i.i.d. is somewhat restrictive; Tullii et al. (2024) have shown that this assumption can be relaxed (under Lipschitz condition on the noise distribution).
- The reliance on uniform price sampling for exploration—while common in the literature—may be impractical when dealing with large feasible price intervals or when the noise distribution’s support is large.

---

> ### Author Response · Authors · 2025-03-10
>
> We thank the reviewer for their valuable feedback and address the points raised below:
>
> - **iid context and integrating Tulli algorithm with antitonic regression.** In the revision we have expanded our discussion regarding the i.i.d. assumption on the context vectors immediately following Assumption 4.2: in our paper, we have worked under a set of standard assumptions used by a number of previous authors.
>
> - **Integration with Tulli et.al (2024) algorithm.** Regarding the integration of the Tulli et.al (2024) algorithm with antitonic regression, one promising approach would be to maintain the Valuation Approximation step from their Algorithm 2 while substituting the Price Elimination step with our antitonic regression procedure (i.e., the blue box in our Algorithm 1). The final optimal price would then be determined as in Equation 5. We acknowledge that this is an initial idea, and further investigation is required to rigorously formalize this integration and conduct a comprehensive regret analysis, which we propose as an interesting direction for future research. In the revision we have added a paragraph on this in the Conclusions section.
>
> - **Non-Linear valuation function.** Incorporating a non-linear valuation function of the form $v_t = g(x_t,\theta_0)+\varepsilon_t$ primarily affects the estimation of $\theta_0$, necessitating a different parametric approach than the one employed in our current algorithm. However, this modification does not introduce any additional challenges in estimating the nonparametric component $F_0$, which continues to be estimated using the antitonic regression procedure as in our algorithm.

---

### Comment · Action_Editor_nsUd · 2025-03-03

Dear Authors,

Thank you for the revision in response to Reviewer enYp's Requested Changes. It would be great if you could respond to the comments of the two other reviewers. We have two weeks before the initial recommendation.

Best,
AE

---

> ### Comment · Action_Editor_nsUd · 2025-03-10
>
> Dear authors,
>
> I would just like to remind you that the discussion period ends in a week. We can start with your responses to the reviews (and also my comment on "On comparison with Tullii at al. (2024)").
>
> Best,
> Junpei

---

### Comment · Action_Editor_nsUd · 2025-04-04
**Clarification on Table 1**

Dear authors,

Regarding the point "Comparison with the work of Tullii et al. (2024)." by reviewer enYp,

The regret bound of Tullii et al. (2024)'s VAPE algorithm (Table 1 in this paper):
* The initial version of this paper state it to be $(dT)^{2/3}$ with probability $1 - 1/(\log T)^{2/3}$,
* the author revised it to $\tilde{O}((dT)^{2/3})$ on Mar 1st,
* and revised it to $\tilde{O}_P((dT)^{2/3})$ on Mar 10th, where $O_p$ holds for some $1 - T^{-a}$ for some $a>0$.

I am wondering which of these versions is the correct information.

FYI: Tullii et al. https://arxiv.org/abs/2406.11316

Best,

AE

---

> ### Author Response · Authors · 2025-04-04
>
> Thank you for your question. The correct information is given in the final version, which aligns with the statement of Theorem 1 in Tullii et al.(2024)'s paper.

---

> > ### Comment · Action_Editor_nsUd · 2025-04-04
> >
> > Dear authors,
> >
> > Thank you for your reply. If I understand correctly, Theorem 1 in Tullii et al.(2024) seems to imply the corresponding regret bound with probability $1 - \tilde{O}(T^{-1})$, which seems to be equivalent to the regret bound of $\tilde{O}((dT)^{2/3}) + T/T \times polylog(T) = \tilde{O}((dT)^{2/3})$ in expectation. Please check this and correct me if I have made a mistake.
> >
> > Best,
> >
> > AE

---

> > > ### Author Response · Authors · 2025-04-05
> > >
> > > Dear AE,
> > >
> > > Thank you for pointing this out. We concur with your claim, assuming that the tail bound on the probability holds for sufficiently large T. However, we remain uncertain whether this bound holds for all sufficiently large T, or only asymptotically as T goes to infinity.
> > >
> > > Best,
> > > the authors

---

> > > > ### Comment · Action_Editor_nsUd · 2025-04-05
> > > >
> > > > Dear Authors,
> > > >
> > > > Thank you for your answer. Could you clarify the difference between "holds for all sufficiently large T" or "only asymptotically as T goes to infinity"? Landau notation is asymptotic and states "there exists a T_0 such that XX holds for all T >= T_0.
> > > >
> > > > Best,
> > > >
> > > > AE

---

> > > > > ### Author Response · Authors · 2025-04-05
> > > > >
> > > > > Dear AE,
> > > > >
> > > > > We have taken a closer look at the updated version of Tulli et al. (2024), and the result is now clear. We agree with your observation that an O(T^{2/3}) bound in expectation can be derived. We have made the corresponding changes in the revised version.
> > > > >
> > > > > Best regards,
> > > > > The authors

---

### Decision · Action_Editor_nsUd · 2025-04-06

**Recommendation:** Accept as is

**Comment:**

Since all three reviewers recommend (lean) acceptance, we are going towards the publication of the paper.

After the revision, all reviewers are positive towards the paper. Reviewer q6TL and S38K called for clarification on the i.i.d. assumption in the context. Reviewer S38K called for clarification on the noise assumption and parameters. Reviewer enYp discusses the comparison with Tullii et al. and clarification on noise assumption. All reviewers are fine with the revision, and the comparison with existing work is now clarified. It seems that no significant issues remain.

**Audience:**

This paper should be interesting to any audience who is interested in contextual dynamic pricing problems. The paper is well-written, and the proposed algorithm is reasonable.

**Claims And Evidence:**

This paper considers online contextual dynamic pricing with linear average valuation and a weak assumption on the noise distribution. At each round, the seller shows a price and the buyer buys it when the price is below his valuation. The paper proposed an epoch-based approach where the size of the epochs doubles for each epoch and each epoch is split into exploration and exploitation parts. The paper derives the performance (regret) bound and conducts simulations with synthetic data as well as more empirical simulations based on real data from Welltower Inc.